# Cell surface protein aggregation triggers endocytosis to maintain plasma membrane proteostasis

David Paul [1], Omer Stern[1], Yvonne Vallis[1], Jatinder Dhillon[2], Andrew Buchanan[2] & Harvey McMahon [1] ✉

The ability of cells to manage consequences of exogenous proteotoxicity is key to cellular homeostasis. While a plethora of well-characterised machinery aids intracellular proteostasis, mechanisms involved in the response to denaturation of extracellular proteins remain elusive. Here we show that aggregation of protein ectodomains triggers their endocytosis via a macro-endocytic route, and subsequent lysosomal degradation. Using ERBB2/HER2-specific antibodies we reveal that their cross-linking ability triggers specific and fast endocytosis of the receptor, independent of clathrin and dynamin. Upon aggregation, canonical clathrin-dependent cargoes are redirected into the aggregation-dependent endocytosis (ADE) pathway. ADE is an actin-driven process, which morphologically resembles macropinocytosis. Physical and chemical stress-induced aggregation of surface proteins also triggers ADE, facilitating their degradation in the lysosome. This study pinpoints aggregation of extracellular domains as a trigger for rapid uptake and lysosomal clearance which besides its proteostatic function has potential implications for the uptake of pathological protein aggregates and antibody-based therapies.

The plasma membrane is constantly changing, as cells regulate surface-exposed proteins in order to adapt their responsiveness to the extracellular environment. One method of regulation is for surface molecules to be endocytosed in response to ligand binding. Endocytic mechanisms can be morphologically divided into those that start with 'ex'vaginations of the plasma membrane and those that start with 'in'vaginations (for review, see ref. [1]). The former, as observed in macropinocytosis and phagocytosis, collectively referred to as macroendocytosis, are characterised by major movements of the plasma membrane towards the extracellular space, and strongly depend on actin polymerisation. Endocytic events starting with invaginations of the cell membrane as observed in clathrin-mediated endocytosis (CME), are typically mediated by binding of adaptor proteins to cytoplasmic receptor tails and frequently depend on dynamin for the scission of

endocytic carriers. Due to constraints imposed by their coat, they result in smaller vesicles.

The two morphologically distinct endocytic routes can be further differentiated by their degree of specificity for endocytic cargo selection. Selectivity in dynamin-dependent forms of endocytosis is believed to be conferred by cytoplasmic cargo adaptors that bind to receptor tails. However, macropinocytic pathways may endocytose a specific region of the membrane but beyond that are thought to be non-selective[2], while phagocytosis is a selective receptor-guided zipper-like process, that locally drives F-actin-induced movement of the membrane around the ingested particle[3].

Dynamics of endocytic events have been followed using fluorescent ligands or antibodies directed against the extracellular domains. The question as to how antibodies accurately reflect or

[1]MRC Laboratory of Molecular Biology, Francis Crick Avenue, Cambridge CB2 0QH, UK. [2]AstraZeneca, R&D BioPharma, Antibody Discovery & Protein Engineering, Granta Park, Cambridge CB21 6GH, UK. ✉e-mail: hmm@mrc-lmb.cam.ac.uk

indeed alter the dynamics of target proteins is of importance not least because of the current use of antibodies as therapeutics to block cell-surface receptor activation, target components of the immune system, or deliver drugs into specific cells[4,5]. Depending which one of these mechanisms is being targeted one may need to optimise/minimise endocytosis and thus the effect of antibodies on the endocytic mechanism of cell-surface receptors needs to be understood.

We started this study using the biparatopic antibody BS4, which binds to and stimulates endocytosis of human epidermal growth factor receptor 2 (HER2, also called ErbB-2/neu)[6,7]. HER2 is over-expressed in 20–25% of breast cancers, associated with aggressive tumour growth and spread, and thus has represented an attractive therapeutic target[8]. The protein is a receptor tyrosine kinase belonging to the epidermal growth factor receptor (EGFR) family. As HER2 is not known to bind any natural ligand it might act on the plasma membrane as a trans-activator by heterodimerising with other receptor family members.

Studying BS4-induced endocytosis of HER2 we find that aggregation may well be a specific selection mechanism that defines molecules to be endocytosed. This aggregation-dependent endocytosis (ADE) is highly reliant on actin polymerisation and can result in the formation of larger vesicles and so may be a form of macro-endocytosis. Antibodies selectively 'activate' this pathway as do other molecules that facilitate cross-linking of cell-surface proteins. ADE responds to aggregation caused by physical and chemical stresses and the resulting pathway leads to intracellular degradation of the aggregates, thus aiding plasma membrane homeostasis. We can envisage protein polymers/aggregates or viruses/micro-organisms with multiple copies of binding proteins also activate the pathway. Thus, an appreciation of this pathway is important for our

understanding of fundamental biological principles as well as therapeutic applications.

## Results

### Efficient endocytosis of biparatopic antibody BS4 does not require clathrin and dynamin but depends on F-actin

Biparatopic HER2-targeting antibody BS4[6] contains the single-chain variable fragment of the human monoclonal antibody, trastuzumab (Tz, brand name Herceptin), N-terminally fused to the heavy chain of 39 S, targeting a distinct HER2 epitope. Thus, the biparatopic antibody has a total of four antigen binding sites (Fig. 1a). Despite similar cell-surface binding, fluorescently labelled BS4 exhibited dramatically increased endocytosis into SkBr3 cells as compared to parental monotopic antibodies (Fig. 1b, c), as previously reported[6,7]. Efficient quenching of the extracellular surface-bound labelled antibody using a non-membrane permeable reducing agent allowed for specific detection of endocytosed antibody in microscopy and flow cytometry-based assays (Supplementary Fig. 1a). Continuous monitoring of pHrodo-labelled antibody uptake confirmed that only BS4 (and not Tz or 39 S) triggered its own endocytosis in the absence of serum (Supplementary Fig. 1b). Given the fast and efficient cellular uptake of BS4, we sought to define its endocytic route using established genetic tools and chemical inhibitors. Clathrin-mediated endocytosis, a major vesicular uptake route into eukaryotic cells[1], is inhibited by expression of the Adaptor Protein 180 C-terminal domain (AP180ct)[9]. While transient expression of AP180ct efficiently inhibited transferrin endocytosis (a canonical marker of the clathrin pathway), uptake of BS4 did not change and was comparable to non-transfected and GFP expressing cells (Fig. 1d and Supplementary Fig. 1d, e). To investigate a wider range of endocytic events we targeted the membrane binding GTPase dynamin, which facilitates membrane fission of clathrin-coated as well as clathrin-

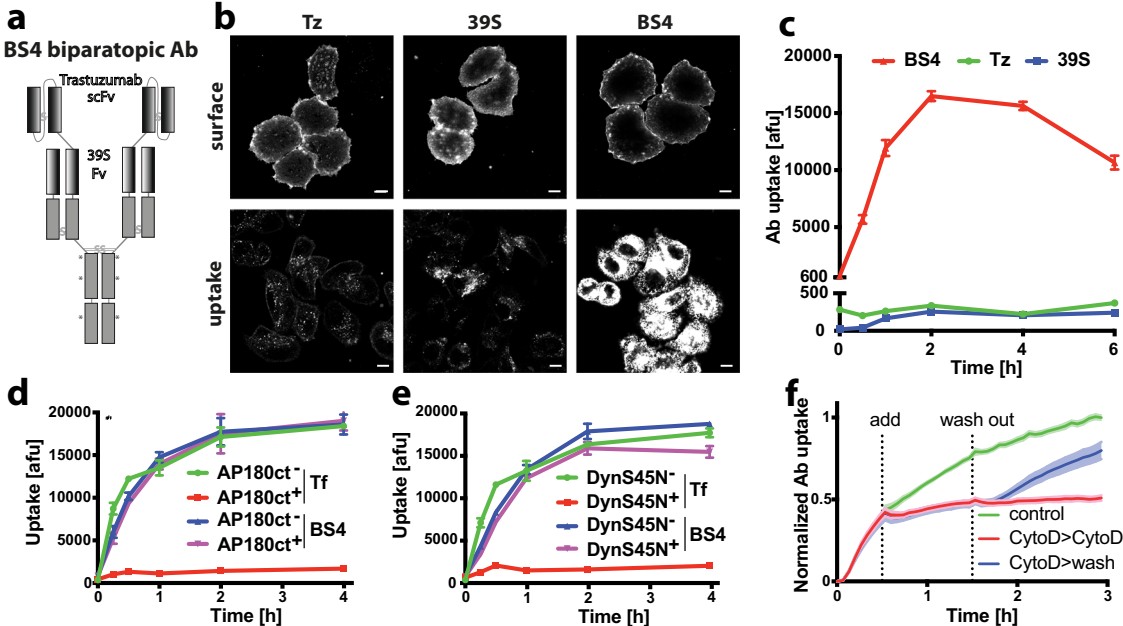

**Fig. 1 | Endocytosis of HER2-specific biparatopic antibody BS4 is not dependent on clathrin and dynamin but requires F-actin. a** Schematic representation of anti-HER2 biparatopic antibody BS4. The Trastuzumab (Tz) single-chain variable fragment (scFv) is attached to the N terminus of the heavy chain of 39 S IgG1 resulting in four antigen binding sites per molecule. Mutations in the Fc region (*) reduce binding to Fc gamma receptor. **b, c** Surface binding and endocytosis of dylight650-labelled monotopic antibodies Tz and 39 S and biparatopic BS4 (all at 3 μg/ml) after 1 h of uptake by confocal microscopy (**b**) and over a period of 6 h by flow cytometry (**c**), (means ± SD, n = 3 independent experiments). **d, e** Endocytosis of BS4 is independent of clathrin and dynamin. SkBr3 cells transfected with dominant-negative

AP180ct (for clathrin-mediated endocytosis) and DynaminS45N N-terminally GFP-tagged expression constructs were incubated with dylight650-labelled BS4 and AlexaFluor546-transferrin for indicated times over a period of 4 h and analysed by flow cytometry. Endocytosis of transferrin and BS4 in cells from the same well expressing or not dominant-negative constructs (AP180ct+ AP180ct-, DynS45N+, DynS45N- in (**d**) and (**e**)), (means ± SD, n = 4 independent experiments). **f** BS4 uptake requires actin polymerisation. SkBr3 cells endocytosing pHrodo-labelled BS4 were treated with 10 μM CytoD, which was subsequently washed out as indicated, (means ± SEM, n = 6 from three independent experiments, two replicate wells each). Scale bars: 10 μm. Source data are provided as a Source Data file.

independent endocytic carriers[10]. Expression of a dominant-negative dynamin-1 S45N (DynS45N, defective for GTP binding)[11,12], potently blocked transferrin uptake, yet BS4 endocytosis remained unaltered as compared to control cells. (Fig. 1e and Supplementary Fig. 1e). Dynamin independence of BS4 uptake was further corroborated using the chemical inhibitor dynasore (Supplementary Fig. 1f). Next we employed a panel of drugs targeting plasma membrane-related processes and found that 5-(N-ethyl-N-isopropyl)amiloride (EIPA) as well as actin polymerisation inhibitor CytochalasinD (CytoD) inhibited BS4 uptake (Supplementary Fig. 1f). EIPA targets the plasma membrane Na$^+$/H$^+$ exchanger and is a common tool to stop macropinocytosis[13]. EIPA inhibited BS4 uptake in a dose-dependent manner (Supplementary Fig. 2a) but showed little specificity over clathrin-mediated endocytosis, as it also inhibited uptake of transferrin into SkBr3 cells (Supplementary Fig. 2a, b). Nevertheless, in HeLa cells ectopically expressing HER2, we determined a concentration range of EIPA that displayed a significant inhibition of BS4 but not transferrin uptake (Supplementary Fig. 2c, d), suggesting that specific inhibition of BS4 endocytosis by EIPA occurs in others cell lines but not SkBr3. On the other hand, inhibition of actin polymerisation by CytoD blocked BS4 uptake in a specific (Supplementary Fig. 2e–g), dose-dependent (Supplementary Fig. 2h), as well as in a fast-acting and reversible manner (Fig. 1f) in all cell lines tested. In addition, the requirement of actin polymerisation for BS4 endocytosis was validated using an Arp2/3 inhibitor (Supplementary Fig. 2i), the removal of which boosted antibody uptake (Supplementary Fig. 2j).

## BS4 uptake morphologically resembles macroendocytosis

The clear clathrin and dynamin independence of BS4 uptake, and the strong requirement for actin polymerisation prompted us to investigate plasma membrane morphodynamics during antibody endocytosis by live-cell microscopy. Fluorescently labelled BS4 bound to the cell surface almost instantly, concentrated into bright fluorescent spots within 2–5 min and localised to endocytic carriers, some with clearly discernible lumina (thus larger than 200 nm in diameter) as early as 7.5 min after antibody addition (Supplementary Movie 1 and Supplementary Fig. 3a). Uptake of 70 kDa dextran (a form of dextran that does not readily label clathrin-coated vesicles) was stimulated in the presence of BS4 (Fig. 2a, b), indicating an increase of fluid-phase uptake, as observed during macropinocytosis[14], upon exposure of cells to BS4. Dextran was localised to the lumina of antibody-positive endocytic vesicles, suggesting co-uptake (Fig. 2c). While binding of BS4 did not induce any alteration in plasma membrane movements (Supplementary Movie 2), membrane-bound antibody was rapidly concentrated by motile lamellipodia and wave-like movements towards the cell body (Supplementary Movies 3 and 4 and Fig. 2d). Membrane lamellipodia that re-extended towards the cell periphery were now depleted of bound antibody. Consistent with the above observations, BS4 was preferentially endocytosed at regions of the cell that exhibited high plasma membrane motility (Supplementary Movie 2 and Supplementary Fig. 3b). Addition of the antibody resulted in the accumulation of the small guanosine triphosphate phosphohydrolase (GTPase) Ras-related C3 botulinum toxin substrate 1 (Rac1) to concentrated spots of BS4 on the cell surface (Fig. 2e and Supplementary Fig. 3c). Abrogation of Rac1-mediated, actin polymerisation-induced lamellipodia movement by a chemical inhibitor (Supplementary Movie 5 and Supplementary Fig. 3d), as well as expression of the dominant-negative T17N Rac1 mutant inhibited BS4 endocytosis (Fig. 2f, g and Supplementary Fig. 3e, f). The strong fluorescence of lamellipodia-clustered BS4 complicated the visualisation of individual endocytic events, so we focussed on more peripheral regions. Indeed, BS4 localised to motile regions of plasma membrane that extended into the extracellular space (Supplementary Movies 6 and 7, Fig. 2h and Supplementary Fig. 3g). These stretches of plasma membrane of several micrometers in length,

subsequently rounded up and folded back onto themselves. Sealing of these cup-shaped structures, concomitant with a movement towards the cytoplasm, gave rise to endocytic vesicles that contained the antibody. We did not observe the uptake of BS4 by membrane invaginations, in agreement with the dynamin independence of BS4 endocytosis. Instead, morphological features of macroendocytosis were evident in the initial membrane movement towards the extracellular space followed by formation of cup-like structures giving rise to large endocytic carriers.

## Aggregation is the trigger for BS4-induced HER2 endocytosis

Next, we determined the molecular features underlying the efficient, self-triggered uptake of BS4. While its cellular ligand, the HER2 receptor exhibited a rather even distribution throughout the plasma membrane in untreated SkBr3 cells, the addition of BS4 rapidly induced HER2 aggregation into large patches followed by rapid internalisation (Fig. 3a). Even though monotopic antibody 39 S efficiently bound to surface HER2 receptor, it did not induce the dramatic redistribution into antibody-receptor clusters. BS4-induced HER2 aggregates in cells were insoluble in a 0.2% Tx100 buffer (Fig. 3b), consistent with previously reported formation of high molecular weight complexes of BS4 and soluble HER2 ectodomain in vitro[6]. Time-dependent aggregation resulted in a concomitant loss in soluble HER2 (Fig. 3b), and was followed by endocytosis of the receptor with an observed delay (Fig. 3b, c). Notably, BS4 aggregated >50% of HER2 within 10 min, with little further increase after longer incubation times (Supplementary Fig. 4a, b). BS4-induced endocytosis of HER2, but not aggregation, was blocked by addition of CytoD. Thus, both microscopy and biochemical analysis suggested that the biparatopic antibody-induced HER2 aggregation preceded the actin-dependent endocytosis of antibody-receptor complexes. We refer to this as aggregation-dependent endocytosis (ADE).

Given the biparatopic nature of BS4 we next ask whether targeting of two distinct epitopes within HER2 was required for receptor aggregation and endocytosis. To this end, we expressed either wt or a mutant version of HER2 lacking the Tz-binding site[15] in CHO cells, which do not express endogenous HER2. BS4 and monotopic antibodies Tz and 39 S bound to CHO cells overexpressing full-length wt HER2, but only the biparatopic antibody was efficiently endocytosed (Supplementary Fig. 4c, top row). As expected, cells expressing the Tz-mutant of HER2 lost Tz binding, but retained 39 S and BS4 binding. However, BS4 failed to aggregate and endocytose the Tz-mutant HER2 receptor (Fig. 3d and Supplementary Fig. 4c, bottom row), suggesting that its four antigen binding sites and targeting of two different epitopes is required to induce aggregation-dependent endocytosis.

Using the reciprocal approach, we cross-linked the two parental monotopic antibodies Tz and 39 S using anti-human IgG, and this leads to more efficient aggregation (Supplementary Fig. 4e) and uptake (Fig. 3e, f) phenocopying BS4. Indeed, targeting of two distinct epitopes i.e., by using the two different monotopic antibodies resulted in more efficient aggregation and subsequent endocytosis (Fig. 3f and Supplementary Fig. 4d, e). We further noted that aggregation-dependent endocytosis (ADE) triggered by cross-linking antibodies was dependent on receptor cell-surface density and was observed in SkOv3 cells to a lesser but detectable extent as compared to SkBr3 cells (Supplementary Fig. 4f) consistent with approximately five times less surface HER2 in SkOv3 compared to SkBr3 cells[6]. We did not observe ADE in MCF7 cells that only express very low levels of HER2. This is not surprising as mathematically the complex formation rate is proportional to the $x$th power of receptor concentration, with $x$ being the product of the receptors oligomeric state and valence of the binding molecule. Thus, only if one can generate a receptor aggregate with a multivalent ligand then one should be able to trigger endocytosis.

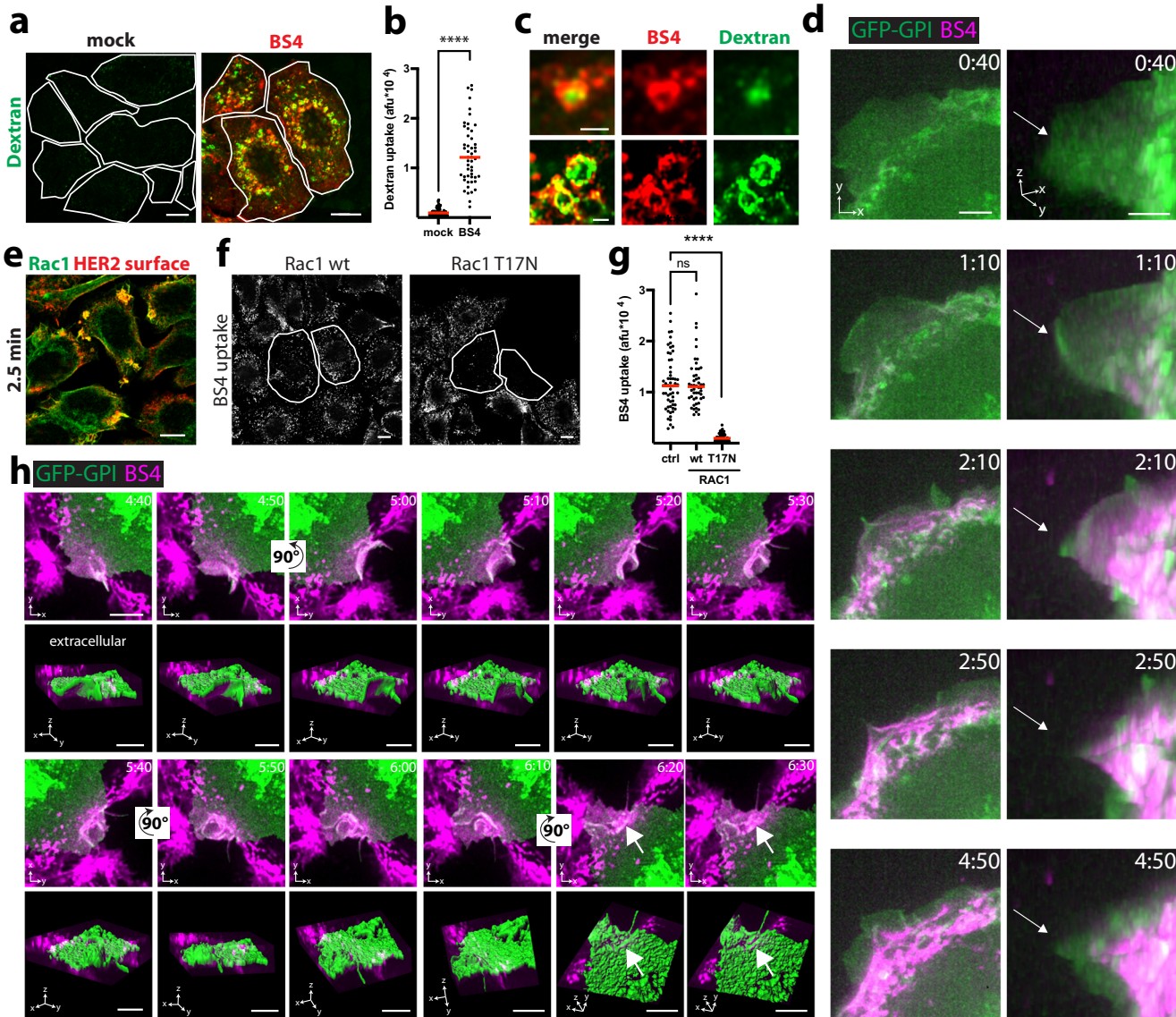

**Fig. 2 | Plasma membrane rearrangements during BS4 uptake are Rac1-dependent and morphologically resemble macroendocytosis.**
**a**, **b** BS4 stimulates fluid-phase uptake. SkBr3 cells were co-incubated with dextran (70 kDa)-TMR and BS4-dylight650 for 10 min and after fixation analysed by confocal microscopy (**a**). The TMR channel is false-colour-coded in green, and individual cells are outlined. **b** Quantification of dextran (70 kDa)-TMR endocytosis in absence (mock) and presence of BS4 (dots represent measurements from individual cells, red lines indicate the median; $n \geq 50$ cells from three independent experiments, ****$P < 0.0001$, two-tailed unpaired Student's $t$ test). **c** BS4-positive endocytic carriers exhibit dextran-filled lumina. **d** Concentration of surface-bound BS4 by inwards moving lamellipodium wave. Cells expressing GPI membrane-bound GFP (green) were imaged immediately after addition of BS4-dylight650 (magenta) (see also Supplementary Movies 3 and 4). 3D views ($x$, $y$, $z$ dimensions) are shown on the right. The arrows indicate the position of the plasma membrane edge at the 40 s timepoint. **e** Rac1 re-localises to BS4-induced HER2 cell-surface aggregates. SkBr3 cells were incubated with BS4 for 2.5 min, fixed, stained for surface HER2 (red) and endogenous RAC (green) and analysed by confocal

microscopy. BS4 endocytosis depends on Rac1 GTPase activity (**f**, **g**). SkBr3 cells transfected with Rac1 wt or dominant-negative T17N C-terminally GFP-tagged expression constructs were incubated with dylight650-labelled BS4 for 10 min. Samples were fixed, surface-bound BS4 was counterstained and cell sections analysed by confocal microscopy (**f**). Subtraction of surface from total BS4 signal shows endocytosed pool (BS4 uptake), transfected cells are outlined. Results are quantified in panel **g**, dots representing measurements from individual cells, red lines indicate the median; $n \geq 50$ cells from three independent experiments, ns (non-significant) $P > 0.05$, ****$P < 0.0001$; one-way ANOVA with Dunnett's multiple comparison test. BS4 is internalised via macroendocytic cups (**h**). SkBr3 cells expressing GFP-GPI (green) were imaged after addition of BS4-dylight650 (magenta) (see also Supplementary Movie 6). Maximum intensity projections are shown and 3D views with surface rendering of the plasma membrane (green) below each panel (see also Supplementary Movie 7). The arrows indicate the concentration of BS4 in endocytic vesicles after internalisation. Scale bars: 10 μm (**a**, **e**, **f**), 1 μm (**c**), 5 μm (**d**, **h**). Source data are provided as a Source Data file.

## Aggregation of transferrin receptor switches its endocytic pathway to ADE, this being induced by cross-linking molecules

To test whether ADE was only induced by HER2-targeting antibodies or rather a more general phenomenon, we next studied endocytosis of TfR, which is a well-established canonical clathrin-dependent receptor[16]. We employed monotopic antibodies targeting the TfR ectodomain, which when combined with a secondary cross-linking

antibody aggregated TfR in vitro (Supplementary Fig. 5a) and in cells (Fig. 4a). Aggregation resulted in increased TfR endocytosis in an F-actin-dependent manner (Fig. 4a–c). Indeed, TfR antibodies were readily endocytosed in CytoD-treated cells, but in the presence of aggregation-inducing secondary antibodies the receptor got completely stuck on the cell surface (Fig. 4b). As expected, expression of dominant-negative AP180ct and DynS45N mutants blocked transferrin

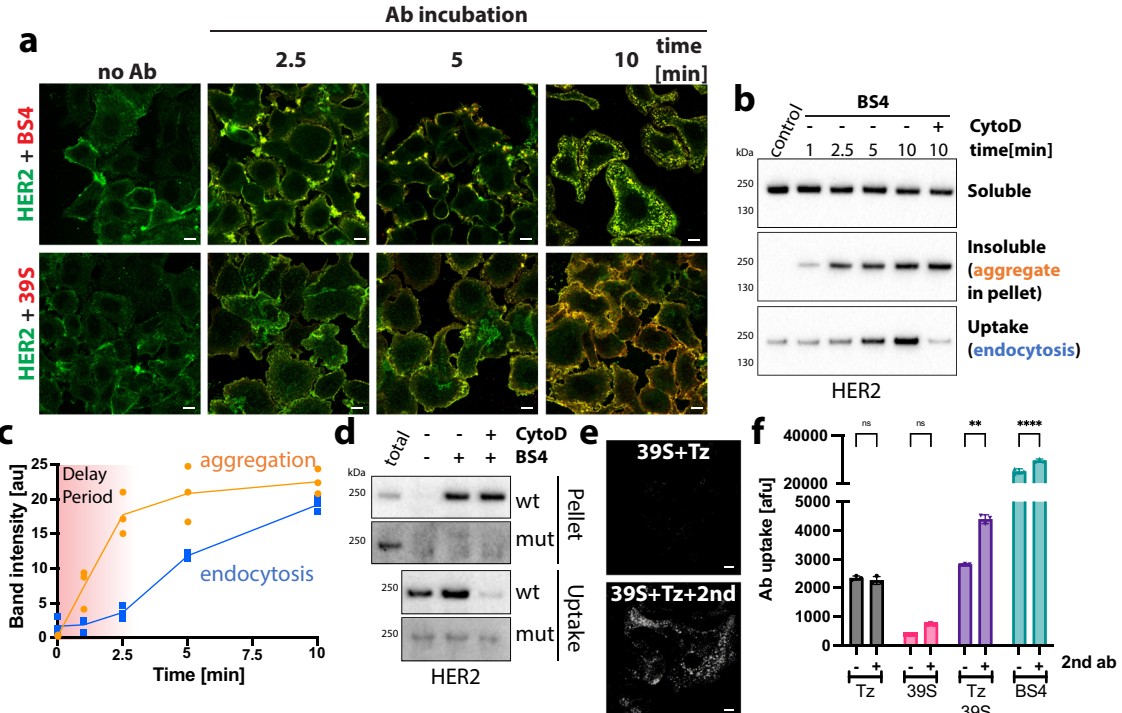

**Fig. 3 | BS4-induced aggregation is the trigger for HER2 endocytosis. a** BS4 clusters HER2 in the plasma membrane prior to endocytosis. SkBr3 cells were incubated with HER2-specific, dylight650-labelled biparatopic antibody BS4 or monotopic antibody 39 S for indicated times. After fixation, total HER2 in cell sections was stained using an antibody against the cytoplasmic domain and samples were analysed by confocal microscopy. **b**, **c** Aggregation of HER2 receptors by BS4 precedes endocytosis. After surface biotinylation, SkBr3 cells were incubated with BS4 for indicated times, surface remaining biotin was removed and samples lysed in low detergent buffer. A fraction was spun to separate soluble (Sup) from insoluble/aggregated proteins (Pellet). Protein in the remaining sample was solubilised (see "Methods" for protocol) and endocytosed biotinylated proteins concentrated on Streptavidin beads (uptake). Samples were assayed by immunoblot for the HER2 protein (**b**). **c** Time dependence of BS4-triggered aggregation and endocytosis of HER2 receptor is quantified (means ± SD, $n = 3$ independent experiments). **d** BS4-triggered cross-linking and endocytosis of HER2 is abrogated for mutant HER2 lacking the Tz-binding site. CHO cells expressing full-length HER2 receptor, either wt or lacking the Tz-binding site, were surface biotinylated and incubated with BS4 for 10 min. Samples were analysed as described in (**b**). **e**, **f** Cross-linking of both monotopic antibodies phenocopies the effect of BS4 on HER2 aggregation and endocytosis. Cells were incubated with equal amounts of indicated antibodies, with or without a cross-linking anti-human Alexa488 antibody (2nd) for 1 h. After fixation surface-bound, dylight650-labelled antibody was quenched and cell sections analysed by confocal microscopy (**e**) or antibody uptake quantified by flow cytometry (**f**) (means ± SD, $n = 3$ independent experiments, ns (non-significant) $P > 0.05$, **$P = 0.0025$, ****$P < 0.0001$, two-way ANOVA with Sidak's multiple comparison test). Scale bars: 10 µm. Source data are provided as a Source Data file.

(Supplementary Fig. 5b) as well as anti-TfR Ab endocytosis (Fig. 4d, e and Supplementary Fig. 5c, d). However, uptake of TfR-antibody aggregates in the presence of a cross-linking secondary antibody occurred independently of clathrin and dynamin (Fig. 4d, e and Supplementary Fig. 5c d). Co-uptake of fluid-phase marker dextran was stimulated upon incubation of cells with TfR antibody only in the presence of the cross-linking secondary antibody (Fig. 4f, g). Dextran localised to the lumina of TfR antibody-positive endocytic vesicles, suggesting co-uptake (Fig. 4h), as also observed during BS4-triggered ADE (Fig. 2c). In addition, endocytosis of TfR antibody in Hela cells was specifically inhibited by EIPA and dominant-negative Rac1 in the presence of a cross-linking secondary antibody (Fig. 4i and Supplementary Fig. 5e, f). This indicated that upon aggregation even clathrin-dependent client receptors are shifted into the ADE pathway, presumably due to cargo size constraints of a clathrin-coated vesicle. Thus, we can redirect a receptor that is normally endocytosed largely by clathrin-mediated endocytosis with antibodies, and points to using caution where labelled antibodies are used as a means of following the endocytic pathways of surface proteins (especially when labelled secondary antibodies are used).

We wondered whether ADE was exclusively triggered by antibodies or whether aggregation by other molecules could trigger the pathway. Therefore, we examined wheat-germ agglutinin, a dimeric lectin, which binds to N-acetyl-D-glucosamine and sialic acid moieties of cell-surface glycoproteins, since it exhibited similar uptake kinetics to BS4 upon binding to the surface of SkBr3 cells (Supplementary Movie 8 and Supplementary Fig. 6a). WGA specifically aggregated glycosylated proteins in vitro (Supplementary Fig. 6b), as well as a panel of surface receptors in cells (Fig. 5a). Their aggregation correlated with stimulation of receptor endocytosis in an F-actin-dependent manner. Interestingly, we observed an increase in protein aggregates for several receptors in the presence of CytoD, suggesting that the failure to endocytose those complexes lead to larger clusters on the cell surface. Simultaneous inhibition of both clathrin/dynamin-mediated endocytosis and F-actin polymerisation completely prevented WGA uptake (Fig. 5b and Supplementary Fig. 6c). However, WGA endocytosis occurred in cells expressing dominant-negative constructs AP180ct and DynS45N, but was strongly reduced in CytoD-treated cells, consistent with the ADE pathway being a major route of WGA uptake. We further corroborated that ADE uptake of receptors was triggered by cross-linking ligands using biotinylated-Tf and tetravalent streptavidin. Endocytosis of biotin-Tf was blocked in cells expressing dominant-negative constructs AP180ct and DynS45N, but readily occurred if cross-linking streptavidin was present (Supplementary Fig. 6d).

## BS4-triggered ADE is HER2-specific

There is a general assumption that macropinocytic pathways are non-selective[2,17] and given the large size of endocytic vesicles that we observed (Supplementary Fig. 3a, arrowheads), we reasoned that these must arise from considerable pieces of the plasma membrane.

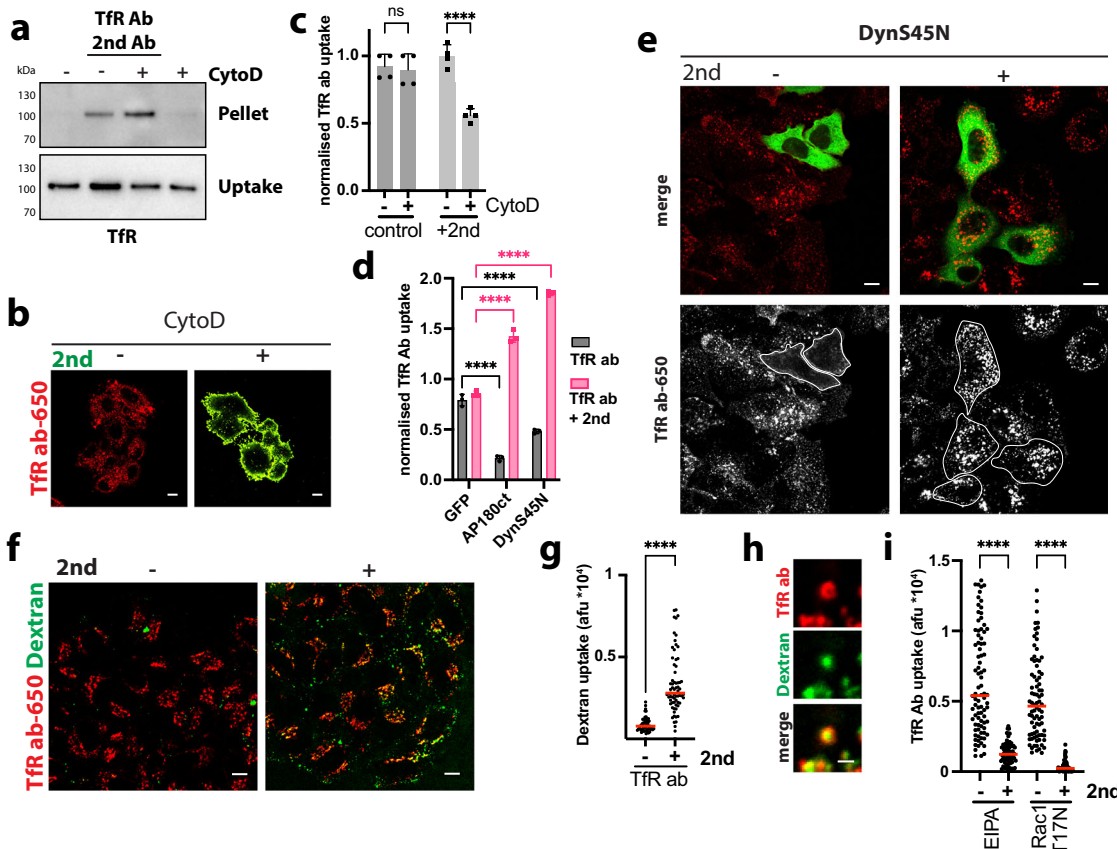

**Fig. 4 | Aggregation-dependent endocytosis re-routes transferrin receptor.**
**a** Antibody-mediated aggregation and endocytosis of TfR. After surface biotinylation, SkBr3 cells ± CytochalasinD (CytoD) were incubated with anti-transferrin-receptor antibody and cross-linking secondary (2nd) antibody for 30 min, surface remaining biotin was removed. Samples were processed, as described in the "Methods" section and assayed by immunoblot for transferrin receptor (TfR).
**b**, **c** Cross-linked transferrin-receptor endocytosis requires actin polymerisation. SkBr3 cells ± CytoD were incubated with dylight650-labelled anti-transferrin-receptor antibody 289 (red) ± secondary antibody-AlexaFluor488 (green) (2nd) for 30 min and analysed by confocal microscopy (**b**) or flow cytometry (**c**) means ± SD, $n = 4$ independent experiments, ns (non-significant) $P > 0.05$, ****$P < 0.0001$, two-way ANOVA with Sidak's multiple comparison test. **d**, **e** Endocytosis of cross-linked transferrin receptor is independent of clathrin and dynamin. SkBr3 cells transfected with control (GFP) and dominant-negative AP180ct and DynaminS45N N-terminally GFP-tagged expression constructs were incubated with dylight650-labelled anti-transferrin-receptor antibody ± secondary antibody (2nd) for 30 min and analysed by flow cytometry (**d**) or confocal microscopy (**e**). Means ± SD, $n = 3$ independent experiments, ****$P < 0.0001$; two-way ANOVA with Dunnett's multiple comparison test shown in (**d**). DynS45N transfected cells are outlined in lower panels (**e**).

**f**, **g** Cross-linked transferrin-receptor endocytosis stimulates fluid-phase uptake. SkBr3 cells were co-incubated with dextran (70 kDa)-TMR (green) and dylight650-labelled anti-transferrin-receptor antibody (red) ± secondary antibody (2nd) for 30 min and analysed by confocal microscopy. Results are quantified in (**g**) dots represent measurements from individual cells, red lines indicate the median; $n \geq 50$ cells from three independent experiments, ****$P < 0.0001$, two-tailed unpaired Student's $t$ test. **h** Cross-linked transferrin-receptor-positive endocytic carriers exhibit dextran-filled lumina. Cross-linked transferrin-receptor endocytosis is inhibited by 5-(N-ethyl-N-isopropyl)amiloride (EIPA) and dominant-negative Rac1 (**i**). HeLa cells transfected with dominant-negative Rac1 (T17N) for 16 h, or treated with 50 μM EIPA for 30 min, were incubated with dylight650-labelled anti-transferrin-receptor antibody ± secondary antibody (2nd) for 30 min. After fixation, surface-bound antibody was counterstained and samples analysed by confocal microscopy. Quantification of anti-transferrin-receptor antibody ± secondary antibody (2nd) endocytosis (after surface subtraction) is shown (dots represent measurements from individual cells, red lines indicate the median; $n \geq 50$ from three independent experiments, ****$P < 0.0001$, one-way ANOVA with Sidak's multiple comparison test). Scale bars: 10 μm (**b**, **e**, **f**), 1 μm (**h**). Source data are provided as a Source Data file.

So we next investigated the selectivity of antibody-triggered ADE. To this end, we monitored the change in surface levels of various plasma membrane localised proteins upon exposure to BS4 in comparison to EGF-stimulated uptake of EGFR. BS4 specifically lowered HER2 surface levels as a consequence of ADE, and the loss of HER2 surface receptors was blocked by CytoD (Fig. 6a, b). Notably, surface receptor levels of EGFR, HER3 and transferrin receptor (TfR) remained unaltered upon BS4 exposure. EGF treatment specifically depleted surface EGFR in an actin polymerisation-independent manner, suggesting that under these conditions EGF-stimulated uptake of EGFR was not occurring by macropinocytosis but more likely by a dynamin-dependent route.

In addition, BS4 specifically accumulated HER2 receptor aggregates into visible patches, which were largely devoid of HER3 and other cell-surface control proteins (Fig. 6c, d and Supplementary Fig. 7).

Biochemical analysis of aggregated antibody-receptor complexes corroborated that BS4 specifically crosslinked HER2 but other surface proteins were not caught in these Tx100-insoluble aggregates (such as HER3, EGFR, Na/K-ATPase and TfR) (Fig. 6e) resulting in a stimulation of HER2 endocytosis and not other proteins monitored (Fig. 6f). This data indicated that similar to clathrin-mediated endocytosis where cargo receptors are concentrated in forming pits[18], BS4-triggered ADE displayed a surprising specificity, achieved by exclusive uptake of membrane patches containing cross-linked receptors, but being largely depleted of other plasma membrane proteins likely by active exclusion during initial formation of extracellular aggregates. Thus, we would argue that specificity is at least partially generated by steric exclusion. We next investigate whether the cytoplasmic tails of aggregated receptors play a role in their endocytosis (and thus could play a role in specificity).

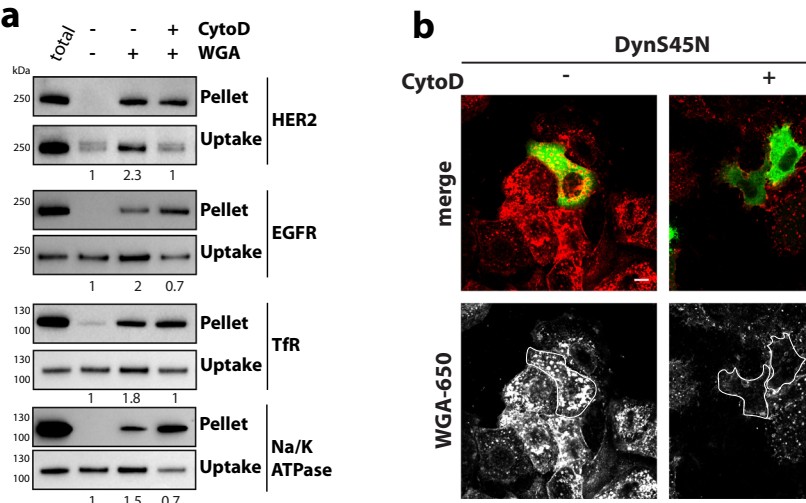

**Fig. 5 | Aggregation-dependent endocytosis is induced by cross-linking molecules. a** WGA aggregates and induces endocytosis of glycosylated proteins in cells. After surface biotinylation, SkBr3 cells ± CytochalasinD (CytoD) were incubated with WGA for 10 min, and remaining surface biotin was removed. Samples were harvested after processing, as described in the "Methods" section, assayed by immunoblot for receptor tyrosine-protein kinase erbB-2 (HER2), epidermal growth factor receptor (EGFR), sodium/potassium-transporting ATPase Alpha1 (Na/K- ATPase) and transferrin-receptor 1 (TfR). Numbers indicate relative levels of uptake normalised to the control sample. **b** WGA uptake is mediated by both clathrin/dynamin-dependent and actin-dependent endocytic pathways. Cells transfected with dominant-negative DynaminS45N N-terminally GFP-tagged expression construct were incubated ± CytochalasinD (CytoD) with dylight650-labelled WGA and analysed by confocal microscopy. Scale bars: 10 μm. Source data are provided as a Source Data file.

## ADE of HER2 is independent of Ras/PI3K but critically requires VAV

High concentrations of growth factors are known to activate membrane ruffling and macropinocytosis via local activation of Rac1-induced actin polymerisation[19]. It is generally believed that ligand-triggered autophosphorylation of tyrosine residues in the cytoplasmic domains of growth factor receptors' initiate a signalling cascade, involving Ras, phosphatidylinositol 3-kinase (PI3K) and its product phosphatidylinositol (3,4,5)-trisphosphate (PIP3), leading to Rac1 activation, which in turn facilitates actin polymerisation and macropinocytosis[3,17,19]. HER2 contains a cytoplasmic kinase domain, which undergoes autophosphorylation upon exposure of SkBr3 cells to BS4 (Fig. 7a). Indeed, the cross-linking antibody induced an increase in cellular tyrosine phosphorylation (Supplementary Fig. 8a, b) co-localising with BS4-triggered HER2 cell-surface aggregates (Fig. 7b). Autophosphorylation was efficiently blocked by using the dual-specific HER2/EGFR small-molecule inhibitor lapatinib[20] (Fig. 7a and Supplementary Fig. 8a, b). Notably, ectopic expression of HER2 lacking the cytoplasmic kinase domain (HER2 ΔCT) exhibited an overall reduction and kinetic defect of BS4 uptake (Supplementary Fig. 8c, d). Yet, in contrast to the full-length protein, HER2ΔCT endocytosed the monotopic antibody efficiently, which was partially independent of actin polymerisation (Supplementary Fig. 8e, f). Thus, we concluded that HER2ΔCT might endocytose constitutively via multiple routes, and that BS4-triggered ADE of HER2 required the cytoplasmic domain for efficient uptake. Given the stimulation of HER2 autophosphorylation upon BS4-induced aggregation, we next assessed the contribution of Ras, PI3K and its lipid product PIP3 to ADE. The addition of BS4 induced a rapid accumulation of PIP3 at plasma membrane ruffles, co-localising with antibody-triggered HER2 aggregates (Fig. 7c and Supplementary Fig. 9a). Both phenomena were efficiently blocked by a PI3K inhibitor. However, while stopping HER2 autophosphorylation reduced BS4 ADE to a similar level as observed for CytoD, inhibition of PI3K/PIP3 had no significant impact on BS4 uptake (Fig. 7d). As a control, we show that Ras-induced macropinocytosis[21] in SkBr3 cells was efficiently blocked by PI3K inhibition (Supplementary Fig. 9b). Consistent with PI3K/PIP3 independence, BS4 was readily taken up into cells expressing a dominant-negative S17N mutant Ras (Fig. 7e, f). Thus,

we concluded that Rac-driven actin polymerisation during BS4-triggered ADE is not activated via Ras-PI3K/PIP3 signalling, but rather an alternative route. Given the central role of guanine nucleotide exchange factors (GEFs) for Rac activation, we next investigated the possible involvement of VAV proteins, which act as RAC-GEFs during phagocytosis[22]. While VAV1 is mostly present in lymphoid and myeloid cells, the other two members of the family VAV2 and VAV3 are more ubiquitously expressed (for protein atlas expression data, see Supplementary Fig. 9c)[23]. Indeed, VAV2 protein was readily detected in SkBr3 cells (Supplementary Fig. 9d) and endogenous as well as over-expressed VAV2 protein was recruited to BS4-induced cell-surface aggregates (Fig. 8g and Supplementary Fig. 9e). Besides their N-terminal actin binding and catalytic (GEF) domain, VAV proteins contain several interaction domains in their C-terminal half, including a pleckstrin homology (PH) domain binding to phosphatidylinositol lipids such as PIP3, an SH2 and two SH3 domain binding to phosphotyrosine residues and proline-rich regions, respectively. VAV2 was recruited to and interacted with BS4-induced HER2 aggregates in a phosphotyrosine-dependent manner (Fig. 8a and Supplementary Fig. 9e), mediated by the SH2 domain (Fig. 8b and Supplementary Fig. 9f). To test a functional role of VAV proteins in BS4-triggered ADE, we expressed dominant-negative mutants (lacking GEF activity), which blocked BS4 uptake (Fig. 8c, d), but not RAS-induced macropinocytosis (Supplementary Fig. 9b). Accordingly, VAV1-3 knockout (KO) cells (Supplementary Fig. 10a, b) displayed a significant reduction in BS4-induced ADE (Fig. 8e, f) and instead large patches of Ab-HER2 aggregates on the cell surface were observed (Fig. 8e, top right panel). Notably, VAV-dependence of HER2 ADE was not restricted to SkBr3 cells but also observed in U2OS cells ectopically expressing full-length HER2 (Supplementary Fig. 10c–f).

Re-expression of any one of VAV1-3 (Supplementary Fig. 10g) rescued the BS4 uptake defect (Fig. 8g) with the extent of rescue correlating with the expression level in individual cells. Additionally, restoration of ADE in VAV1-3 KO cells required re-expression of a VAV protein containing the SH2 domain (Fig. 8h and Supplementary Fig. 10h), confirming the HER2 phosphotyrosine - VAV-SH2 domain-dependent recruitment of VAV-facilitated ADE.

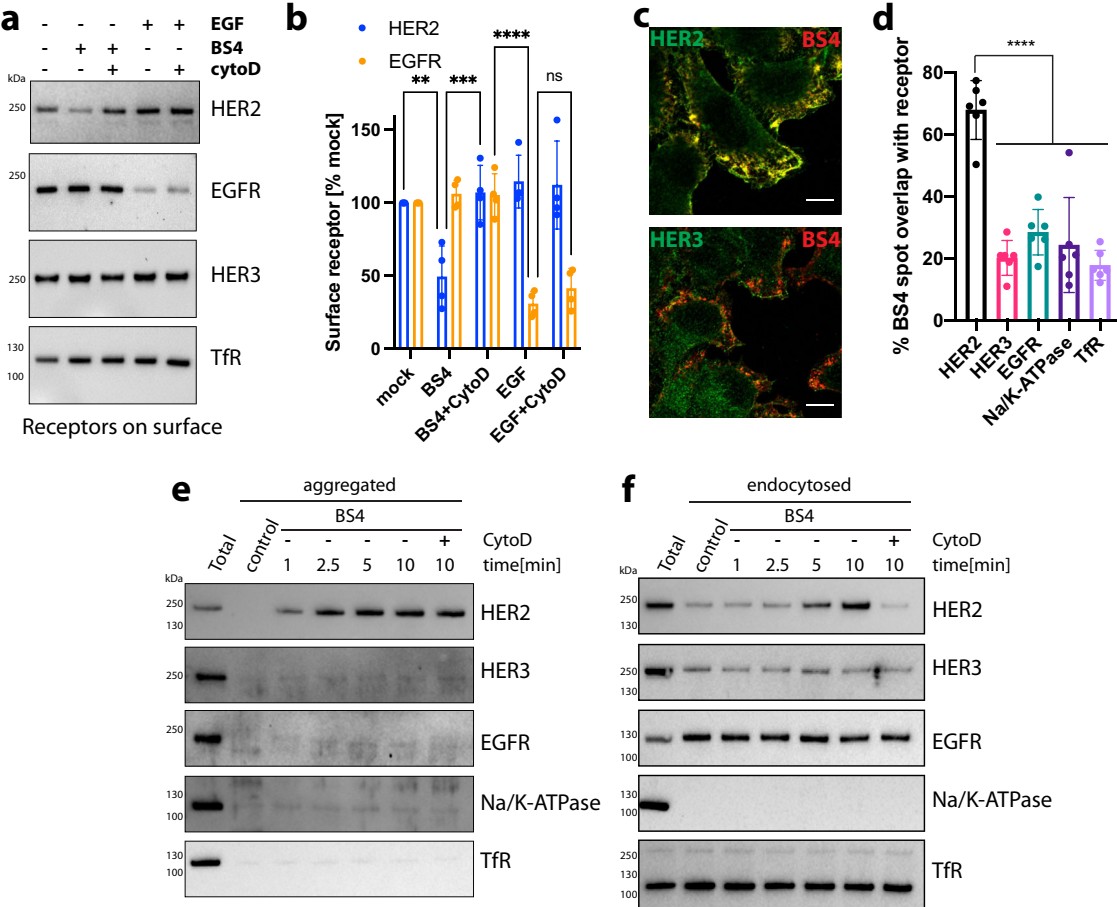

**Fig. 6 | BS4-triggered aggregation and actin-dependent macroendocytosis is HER2-specific. a**, **b** Depletion of HER2 and EGFR surface receptor levels by treatment with BS4 and EGF, respectively. SkBr3 cells were incubated for 10 min with BS4 or EGF, and ± CytochalasinD (CytoD) followed by cell-surface biotinylation and concentration on Streptavidin beads (surface). Samples were analysed by immunoblot for HER2, EGFR and HER3, TfR as negative controls (**a**). A quantitation of the results is shown in (**b**) means ± SD, $n = 4$ independent experiments, ns (non-significant) $P > 0.05$, **$P = 0.0051$, ***$P = 0.0009$, ***$P < 0.0001$; two-way ANOVA with Sidak's multiple comparison test. **c**, **d** BS4 clusters HER2 but no other receptors in the plasma membrane resulting in HER2-specific endocytosis. SkBr3 cells were incubated with HER2-specific, dylight650-labelled biparatopic antibody BS4 for 2.5 min. After fixation, total receptor tyrosine-protein kinase ErbB-2 (HER2), receptor tyrosine-protein kinase ErbB-3 (HER3), epidermal growth factor receptor (EGFR), sodium/potassium-transporting ATPase Alpha1 (Na/K-ATPase) and

Transferrin-receptor 1 (TfR) in cell sections were stained and samples were analysed by confocal microscopy. The spot overlap of fluorescent signal for BS4 and respective receptors is quantified in (**d**); means ± SD, $n = 6$ randomly chosen fields of view with a total of at least 50 cells per condition, ***$P < 0.001$, one-way ANOVA with Dunnett's multiple comparison test. **e** Specific aggregation of HER2 receptor by BS4. SkBr3 cells were incubated with BS4 for indicated times and samples lysed in the low detergent buffer. Samples were spun to separate soluble (Sup) from insoluble/aggregated proteins (Pellet) and immunoblotted for indicated cell-surface receptors. **f** BS4 specifically endocytoses HER2 receptor. After surface biotinylation, cells were incubated with BS4 for indicated times. Protein was fully solubilised (see "Methods" for protocol) and endocytosed biotinylated proteins concentrated on Streptavidin beads (uptake). Samples were assayed by immunoblot for the HER2, HER3, EGFR, Na/K-ATPase and TfR. Scale bars: 10 μm. Source data are provided as a Source Data file.

## ADE clears stress-induced receptor aggregates to aid cell surface proteostasis

Finally, we asked whether ADE was exclusively triggered by cross-linking ligands, or if the pathway was also activated upon exposure of plasma membrane receptors to physical or chemical stress, resulting in surface localised protein aggregates. While a plethora of well-characterised machinery aids intracellular proteostasis, mechanisms involved in response to alterations within ectodomains of plasma membrane proteins remain largely elusive. We screened several chemical and physical perturbations and found that exposure to low pH as well as heat-shock treatment triggered HER2 aggregation followed by F-actin-dependent endocytosis in SkBr3 cells (Fig. 9a). The extent of ADE efficiency correlated with the amount of stress-induced HER2 aggregation on exposure to various pHs (Supplementary Fig. 11a) as well as different temperatures heat shocks (Supplementary Fig. 11b). Upon stress-induced ADE, HER2 was found in large endocytic vesicles (Fig. 9b, insets) independent of dynamin

(Fig. 9b) and clathrin (Supplementary Fig. 11c), but inhibited by CytoD treatment (Fig. 9b). Notably, uptake of fluid-phase marker dextran was stimulated by heat stress-induced ADE and was inhibited by EIPA and CytoD treatment (Fig. 9c, d). Indeed, we observed stress-induced uptake via ADE of HER2, TfR and EGFR (Supplementary Fig. 11d, e), arguing that in contrast to Ab/ligand-induced ADE which results in the specific uptake of a particular cognate Ab-receptor complex, stress-induced ADE facilitates the uptake of all aggregated (Supplementary Fig. 11e, Pellet) receptors from the cell surface. While both scenarios highlight the cell-surface aggregate as the trigger for endocytosis, the plethora of stress-induced cell-surface receptor aggregates was consistent with multiple signalling pathways contributing to actin-driven ADE in this case. We observed a partial reduction of heat-shock-triggered ADE upon depletion of VAV proteins, inhibition of EGFR/HER2-mediated tyrosine phosphorylation or blocking PI3K activity, while concomitant inhibition completely abrogated heat-shock induced dextran uptake (Supplementary

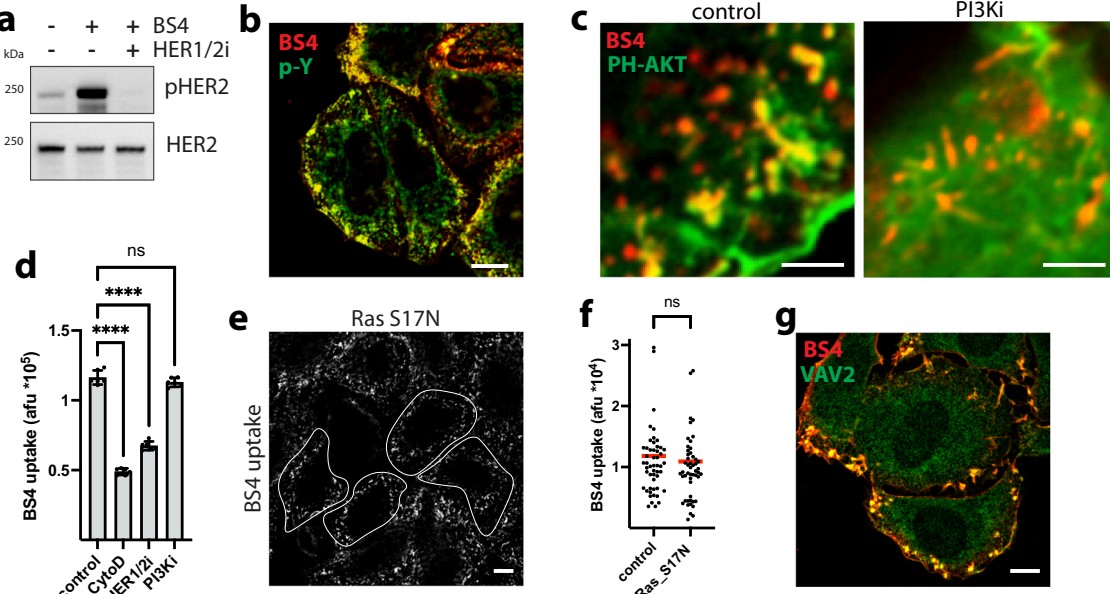

**Fig. 7 | BS4-triggered ADE of HER2 is independent of Ras and PI3K.** BS4 induces HER2 autophosphorylation (**a, b**). SkBr3 cells ± dual EGFR/HER2 kinase inhibitor lapatinib (HER1/2i) were incubated with BS4 for 2.5 min as indicated. Samples were analysed by immunoblot for total HER2 and phospho-HER2 (pHER2) (**a**). SkBr3 cells were incubated with BS4-dylight650 (red) for 2.5 min, after fixation stained for phosphorylated tyrosine residues (p-Y, green) and analysed by confocal microscopy (**b**). Localisation of PIP3 to BS4-induced HER2 aggregates (**c**). SkBr3 cells were transfected with a vector expressing the PIP3 sensor PH-AKT-GFP (green) for 16 h treated or not with the PI3K inhibitor LY294002 (PI3Ki) followed by incubation with BS4-dylight650 (red) for 2.5 min. After fixation samples were analysed by confocal microscopy. BS4 uptake is reduced by HER2 kinase but not PI3-kinase inhibition (**d**). SkBr3 cells were treated with CytoD, lapatinib (HER1/2i) or LY294002 (PI3Ki) as indicated and incubated with BS4-dylight650 for 30 min. After fixation BS4 uptake was quantified by flow cytometry; means ± SD, $n = 6$ independent experiments, ns

(non-significant) $P = 0.252$, ****$P < 0.0001$; one-way ANOVA with Dunnett's multiple comparison test. BS4-induced ADE is independent of Ras (**e, f**). SkBr3 cells were transfected with a plasmid expressing a dominant-negative mutant (S17N) of Ras for 16 h followed by incubation with BS4-dylight650 for 30 min. After fixation surface-bound BS4 was counterstained and samples analysed by confocal microscopy. Subtraction of surface from total antibody signal yielded the endocytosed pool (**e**) with transfected cell outlined. Results are quantified in (**f**); dots represent measurements from individual cells, red lines indicate the median; $n \geq 50$ cells from three independent experiments, ns (non-significant) $P = 0.532$; two-tailed unpaired Student's $t$ test. Localisation of VAV2 to BS4-induced HER2 aggregates (**g**). SkBr3 cells were incubated with BS4-dylight650 (red) for 2.5 min. After fixation, samples were stained for endogenous VAV2 and analysed by confocal microscopy. Scale bars: 10 μm (**b, e, g**), 3 μm (**c**). Source data are provided as a Source Data file.

Fig. 12a–d). Notably, as opposed to BS4-mediated endocytosis of HER2, ADE of antibody-induced TfR aggregates was Ras and PI3K kinase-dependent, but did not require VAV proteins (Supplementary Fig. 12e–g). These results supported the notion that while ADE generally requires actin-driven membrane rearrangements, the activation pathways for actin polymerisation may well differ and will be determined by the aggregated receptor.

Next, we investigated the fate of receptor aggregates after ADE. BS4 treatment induced the time-dependent degradation of Ab-HER2 receptor complexes, while monotopic antibodies did not induce strong alterations in steady-state HER2 protein levels (Fig. 10a and Supplementary Fig. 13a). Since intracellular degradation of HER2 was mediated by acid-dependent proteases (Supplementary Fig. 13b), we employed V-ATPase inhibitor BafilomycinA1 (BafA1) and found inhibition of BS4-induced degradation of HER2 (Supplementary Fig. 13c) (but not uptake of Ab-receptor complexes) (Supplementary Fig. 13d, e) was blocked by inhibition of lysosomal acidification. Consistent with the kinetics and low pH-dependence of HER2 degradation, BS4 was found in lysosomal compartments staining positive for LAMP-1 (Supplementary Fig. 13f, g). Internalisation of stress-induced receptor aggregates also resulted in intracellular degradation (Fig. 10b, c), which was blocked by inhibition of ADE (CytoD) as well as lysosomal acidification (BafA1). Endocytosed receptor aggregates co-localised with lysosomal marker LAMP-1 (Supplementary Fig. 13h) in a time-dependent manner. This data suggests that stress-induced receptor aggregates are quickly cleared from the cells surface by ADE and degraded inside the lysosome, thereby facilitating plasma membrane homeostasis. Lastly, we wondered about the physiological impact on

cell fitness of ADE-mediated cell-surface receptor aggregate clearance. We thus aggregated HER2 using BS4 in SkBr3 cells inhibited for ADE and monitored the effect on cell growth. While BS4 did not impact cell growth when ADE was operational, the accumulation of BS4-induced HER2 surface aggregates in the presence of CytoD negatively impacted cell fitness (Fig. 10d). This effect was due to extracellular HER2 aggregates and not observed with monotopic control antibody Tz, which binds but does not cluster HER2. Thus, rapid removal of cell-surface aggregates is needed for cell homeostasis.

## Discussion

The concept that antibodies can cross-link receptors and cause patching and capping on the plasma membrane has long been recognised[24–26]. Furthermore, receptor cross-linking is an established way for lysosomal targeting[27], however with limited and contradictory information on the endocytic pathway. While some studies suggest that endocytosis is triggered by antibodies, often with the assumption that the receptor is the main determinant of the endocytic pathway[28], our data show that cross-linking antibodies redirects receptors into ADE, in line with what has been proposed for the uptake of Ab-aggregated ricin toxin endocytosis[29]. Thus, studies which use antibodies to follow protein dynamics, in particular those that use labelled secondary antibodies will need to be re-evaluated in the light of our observations, as the combination of primary and secondary antibodies will likely lead to cross-linking of target proteins and stimulate ADE.

Antibodies are invaluable diagnostics and therapeutics tools. However, when targeting cell-surface proteins, the design of therapy

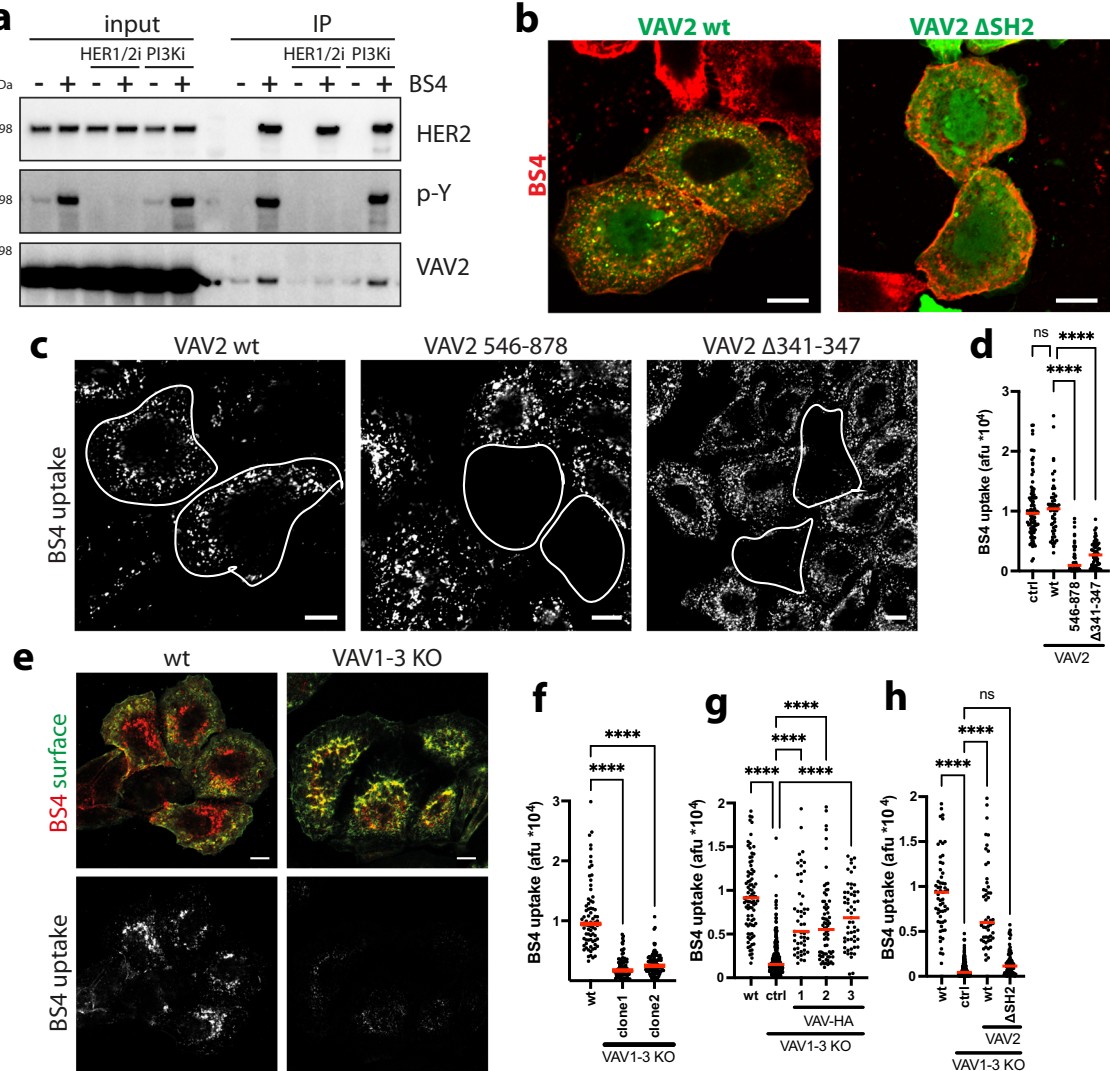

**Fig. 8 | BS4-triggered ADE of HER2 critically requires VAV.** BS4-induced inter-action of HER2 with VAV2 is phosphotyrosine dependent (**a**). SkBr3 cells ± lapatinib (HER1/2i) or LY294002 (PI3Ki) were incubated with BS4 for 2.5 min. BS4-HER2 aggregates were immunoprecipitated and samples analysed by western blot. VAV2 is recruited to BS4-HER2 aggregates via its SH2 domain (**b**). SkBr3 cells were transfected with VAV2 wt or lacking residues 665–772 (ΔSH2) C-terminally GFP-tagged expression vector (green) for 16 h followed by incubation with BS4-dylight650 (red) for 10 min. After fixation samples were analysed by confocal microscopy. BS4-induced ADE requires VAV2 GEF activity (**c**, **d**). SkBr3 cells were transfected with VAV2 wt, comprising amino acids 546–878, or lacking catalytic residues (Δ341–347) for 16 h followed by incubation with BS4-dylight650 (red) for 30 min. After fixation, surface-bound BS4 antibody was counterstained and sam-ples were analysed by confocal microscopy. Subtraction of surface from total BS4 signal shows an endocytosed pool (BS4 uptake), transfected cells are outlined. Results are quantified in (**d**). VAV proteins are required for BS4 uptake (**e**, **f**). SkBr3 wt or VAV1-3 knockout (KO) cells were incubated with BS4-dylight650 (red) for 30 min. After fixation, the surface-bound BS4 antibody was counterstained (green) and samples were analysed by confocal microscopy (**e**). Subtraction of surface from total BS4 signal shows endocytosed pool (BS4 uptake). Results are quantified in (**f**). Rescue of BS4 uptake in SkBr3 VAV1-3 knockout cells by expression of VAV1-3 (**g**). SkBr3 wt or VAV1-3 knockout (KO) cells were transfected with C-terminally HA-tagged VAV1-3 expression vectors for 16 h followed by incubation with BS4-dylight650 (red) for 30 min. Samples were processed, analysed and BS4 uptake quantified as described in (**e**). Rescue of BS4 uptake in SkBr3 VAV1-3 knockout cells depends on the SH2 domain in VAV2 (**h**). SkBr3 wt or VAV1-3 knockout (KO) cells were transfected with VAV2 wt or a lacking residue 665–772 (ΔSH2) C-terminally GFP-tagged expression vectors for 16 h followed by incubation with BS4-dylight650 for 30 min. Samples were processed, analysed and BS4 uptake quantified as described in (**e**). Quantifications in (**d**, **f**, **g**, **h**): dots represent measurements from individual cells, red lines indicate the median; $n \geq 50$ cells from three independent experiments, ns (non-significant) $P > 0.05$, ****$P < 0.0001$; one-way ANOVA with Dunnett's multiple comparison test. Scale bars: 10 μm (**b**, **c**, **e**). Source data are provided as a Source Data file.

must take into account how the antibodies may affect target endocy-tosis, as this will strongly influence its efficacy. Minimising endocytosis might be warranted when therapeutic intervention aims to block receptor activation or recruit immune cells. On the other hand, anti-bodies purposed to downregulate the surface antigen or deliver a (cytotoxic) cargo, will only work if uptake is triggered efficiently. Our work begins to provide the basis of a rationale as to how one can optimise antibodies for specific purposes.

BS4-induced aggregation of HER2 receptors appears to precede their uptake, suggesting that aggregation-dependent autopho-sphorylation provides the cytoplasmic signal for endocytosis.

Such a cytoplasmic signal helps explain the embedded specificity in removing primarily cross-linked receptors. However, also natural ligands as well as non-cross-linking antibodies such as Trastuzumab can trigger HER2 dimerisation-induced autophosphorylation[30,31], but this does not result in ADE. Thus, we envisage that a threshold of tyrosine-

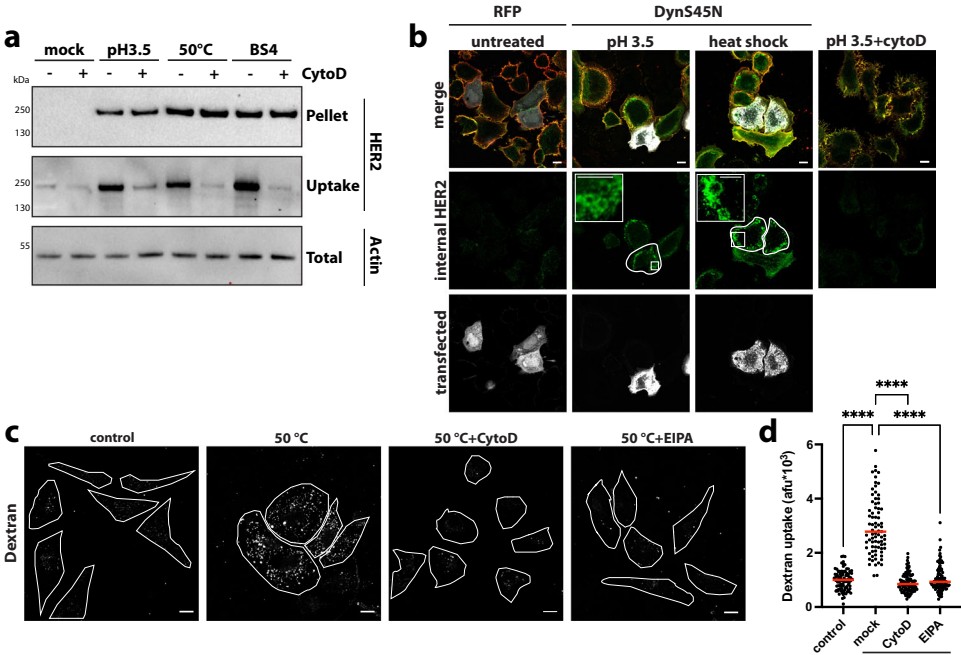

**Fig. 9 | Stress-induced cell-surface receptor aggregates are endocytosed by ADE. a** HER2 receptor aggregation and endocytosis upon chemical and physical stress. After surface biotinylation, SkBr3 cells ± CytochalasinD (CytoD) were washed in an acidic buffer (pH 3.5), heat shocked for 5 min (at 50 °C), or treated with BS4. After incubation of all samples at 37 °C for 30 min, surface remaining biotin was removed and samples lysed in low detergent buffer. A fraction was spun to precipitate insoluble/aggregated proteins (Pellet). Protein in the remaining sample was solubilised (see "Methods" for protocol) and endocytosed biotinylated proteins concentrated on Streptavidin beads (uptake). Samples were assayed by immunoblot HER2. **b** Stress-induced aggregation-dependent endocytosis occurs independent of dynamin but requires actin polymerisation. SkBr3 cells transfected with RFP control, or dominant-negative DynaminS45N N-terminally RFP tagged expression vector (RFP channel displayed in white) were heat shocked for 5 min (at 50 °C) or washed in an acidic buffer (pH 3.5) ± CytochalasinD (CytoD) followed by 30 min incubation at 37 °C. After fixation surface HER2 was stained (false-colour-coded in red) before permeabilisation and staining for total HER2 (green channel). Cell sections were analysed by confocal microscopy and post-processed by subtraction of surface from total HER2 staining, which allowed for specific visualisation of intracellular HER2. **c, d** Heat stress induces fluid-phase uptake. HeLa cells ± CytochalasinD (CytoD) or ± 5-(N-ethyl-N-isopropyl)amiloride (EIPA) were incubated with dextran (70 kDa)-fluorescein incubated for 5 min at 50 °C followed by 30 min at 37 °C. After fixation samples were analysed by confocal microscopy (**c**). Quantification of heat stress-induced dextran uptake is shown in (**d**). Dots represent measurements from individual cells, red lines indicate the median; $n \geq 50$ cells from three independent experiments, ****$P < 0.0001$; one-way ANOVA with Sidaks's multiple comparison test. Scale bars: 10 μm (**b**, **c**), 1 μm insets panel **c**. Source data are provided as a Source Data file.

phosphorylation signal per membrane area has to be overcome to trigger ADE. How the extent of cross-linking will regulate the endocytic response, i.e., what aggregate size/autophosphorylation signal threshold is required to induce ADE, remains to be determined. Also signalling during growth factor-triggered macropinocytosis is initiated by ligand-induced autophosphorylation of tyrosine residues, subsequently involving Ras, PI3K/PIP3 to activate Rac1, which in turn facilitates actin polymerisation and macropinocytosis[3,17,19]. On the contrary, we find BS4-triggered ADE of HER2 to be independent of Ras/PI3K signalling. Instead, the Rac1-GEF VAV is directly recruited in a phospho-tyrosine-dependent manner via its SH2 domain to facilitate Rac-mediated actin polymerisation and ADE, similar to what has been described for Fcχ-receptor-mediated phagocytosis[32]. Thus, ADE of HER2 exhibits striking parallels to phagocytosis as a selective receptor-guided zipper-like process, that locally drives F-actin-induced movement of the membrane around the ingested particle[3], which in the case of BS4-triggered ADE of HER2 is the antibody-receptor aggregate. Even though we find PI3P enrichment at BS4-induced HER2 surface aggregates, pharmacological inhibition of PI3K activity did not block ADE even though PIP3 recruitment to aggregates was stopped. Thus, BS4-triggered ADE does not require PI3K/PIP3 contrary to its previously proposed role in completion of macroendocytic events[33]. Alternatively, PI3K-dependent recruitment of Rac1-GEFs[34] might feed into the activation of actin polymerisation and ADE, which however in the case of BS4 uptake is not required presumably due to its extraordinary capacity to cross-link HER2 receptors. Accordingly, our data show that aggregation of

different receptors results in ADE involving different signalling pathways to activate Rac1. As opposed to BS4-mediated uptake of HER2 requiring VAV proteins, ADE of antibody-induced TfR aggregates was Ras and PI3K kinase-dependent but did not require VAV proteins. Thus, even though ADE relies on actin-driven membrane rearrangements, the activation pathways for actin polymerisation may well differ and will be determined by the aggregated receptor. In addition, we envision that as in the case of stress-induced surface aggregates involving a plethora of cell-surface receptor aggregates, activation of multiple signalling pathways in parallel can contribute to actin-driven ADE. In summary, aggregation of receptors is the signal for their internalisation via a Rac-actin-dependent endocytic pathway and the use of alternative (receptor-dependent) pathways of Rac activation may well reflect the built-in robustness in biology and thus probably highlights the important homoeostatic role of ADE.

There are many well-described mechanisms for cytoplasmic protein quality control. However, information is scarce on how unfolded or damaged surface proteins are sensed and dealt with[35–37]. In yeast, denatured proteins are ubiquitinated by Rsp5, followed by ART adaptor protein binding, endocytosis and vacuolar degradation[37]. This is likely a sensing mechanism for unfolded cytoplasmic domains (as opposed to extracellular domains). Notably, yeast cells are believed to be incapable of macropinocytosis presumably due to the rigid cell wall[38], thus also likely being unable to remove cell-surface aggregates via ADE. Whether Rsp5-ART homologous machinery acts in higher eukaryotes, with the ability to broadly remove aggregated plasma

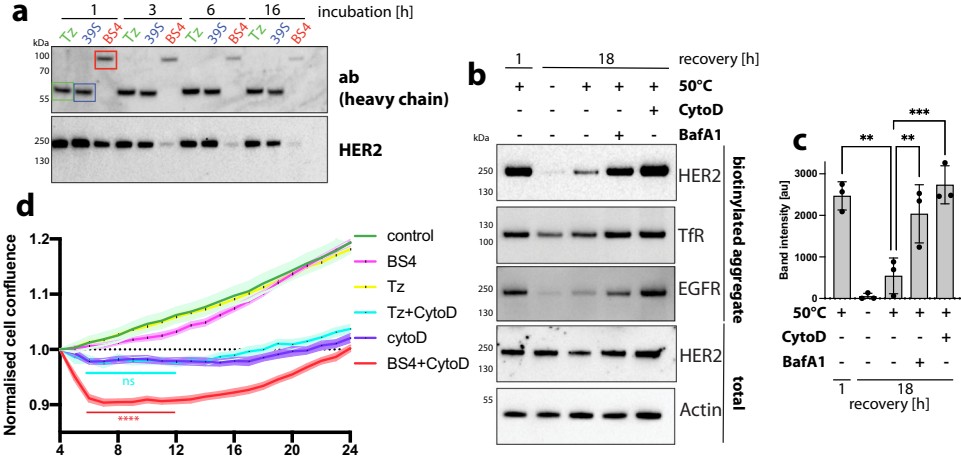

**Fig. 10 | ADE facilitates lysosomal degradation of endocytosed receptor aggregates. a** Time-dependent degradation of endocytosed antibody-receptor aggregates. Western blot analysis of steady-state HER2 protein and antibody levels after exposure to monotopic antibodies Tz and 39 S as well as biparatopic antibody BS4 for increasing length of time. **b, c** Stress-induced aggregates are degraded in the lysosome after endocytosis. SkBr3 cells ± CytochalasinD (CytoD) or Bafilomy-cinA1 (BafA1) were surface biotinylated, heat shocked for 5 min (at 50 °C) and subsequently incubated for the indicated length of time at 37 °C. Samples were lysed in a low detergent buffer, insoluble/aggregated proteins precipitated by centrifugation, biotinylated receptors isolated as described in materials and methods and analysed by immunoblot for HER2, EGFR and TfR. Total cell lysates analysed for HER2 and actin are shown as controls. **c** The amount of intracellular

HER2 aggregates at different times points is quantified. Means ± SD, $n = 3$ independent experiments, $**P ≤ 0.0067$, $***P = 0.0004$; one-way ANOVA with Sidaks's multiple comparison test. **d** Presence of extracellular receptor aggregates negatively affect cell growth. SkBr3 cells were incubated with monotopic antibody Tz or biparatopic antibody BS4 ± CytochalasinD (CytoD) for 4 h at 37 °C. Cell confluence was determined hourly using an Incucyte live-cell imager. Data normalised to the 4-h timepoint, at which antibodies and CytoD were removed, are displayed. The representative result of three independent experiments is shown. means ± SD, $n = 8$ replicate wells, significance shown for CytoD vs CytoD+BS4 (red) and CytoD vs CytoD+Tz (turquoise) 5–12 h time points ns $P > 0.05$, $****P < 0.0001$; two-way ANOVA with Tukey's multiple comparison test. Source data are provided as a Source Data file.

membrane receptors is less clear[39]. One well-studied example in mammals is cystic fibrosis transmembrane conductance regulator (CFTR) a polytopic transmembrane protein, which is endocytosed upon conformational damage recognition of its cytoplasmic tail by chaperones, followed by ubiquitination and ESCRT-mediated degradation in the lysosome[40]. HER2 receptors are stabilised at the plasma membrane by binding of Hsp90 chaperones to their cytoplasmic domain[41]. Upon Hsp90 dissociation (e.g., using chaperone inhibitors) HER2 is ubiquitinated followed by clathrin-mediated endocytosis and proteasomal and/or lysosomal degradation[42–44]. While this may represent a feasible way to endocytose and degrade damaged single receptors, it is unlikely to be operational for large aggregates due to the size constraints of the clathrin-coated vesicle. Nevertheless, BS4 was previously described to trigger ubiquitination of HER after ~30 min of exposure to the antibody[7]. However, we identified fast (within few minutes) HER2 autophosphorylation to be critical for endocytosis via ADE, while its ubiquitination may well be involved in subsequent targeting to the lysosome.

Here, we describe a robust mechanism by which proteins that aggregate will be removed from the cell surface and degraded. We tested for this using a pH insult and heat shock. In both cases, we observed rapid endocytosis that exhibited defining characteristics of macroendocytosis (large vesicles, actin-dependence, clathrin/dynamin independence, co-uptake of dextran, inhibition by EIPA). In line with these insults having general effects on surface protein integrity causing general protein aggregation, there was no apparent specificity to the receptors being endocytosed, but multiple receptors can be found in the same vesicle. Thus we suggest that ADE may be an important mechanism for plasma membrane protein quality control, capable of removing large aggregates as an additional layer of quality control besides the ART-ubiquitin-CME system for the retrieval/removal of single receptors.

Macropinocytosis has been observed across many different species. There are some suggestions that it allows cells to imbibe nutrients

from the milieu and that this may be a reason that cancer cells show evidence of an upregulated macropinocytosis[45]. While this may be the case, we would suggest that potentially a more fundamental reason cells need this pathway in the long term is to keep the cell surface free of aggregated proteins (or large material that bind to the cell surface leading to protein clustering/aggregation), and that this is a matter necessary for health.

## Methods

### Antibodies, ligands and proteins
HER2-specific antibodies Trastuzumab, 39 S, BS4 as well as TfR-specific antibodies 226, 289 and 292 were transiently expressed by co-transfection of the heavy and light chain expressing vectors into CHO cells using a polyethylenimine-based method[46]. Supernatants were purified by affinity chromatography using a Protein A MabS-electSure column (GE Healthcare) and molecules were analysed for integrity by SDS-PAGE and SEC-HPLC using a TSKgel G3000SWxl column (Tosoh Bioscience). Wheat-germ-agglutinin (L9640) and biotin-transferrin (T3915) were from Merck. Antibodies/ligands were labelled with DyLight-650 or 488 NHS Ester (Thermo Fisher Scientific 62265 and 46402) according to manufacturers' instructions. FabFluor Red Antibody Labelling Reagent (4722) was from Sartorius. Transferrin-AlexaFluor546 (T23364) and EGF-Alexa647-complex (E35351), Dextran (70 kDa)-Tetramethylrhodamine (D1818) and Dextran (70 kDa)-Fluorescein (D1822) were from Thermo Fisher Scientific. Antibodies recognising the HER2 cytoplasmic domain (2242), pHER2 (2241), HER3 (12708), the HA-tag (2367) and EGFR (4267) were from Cell Signaling Technology. An antibody recognising the HER2 ectodomain (AF1129) was from R&D Systems. Antibodies recognising VAV1 (ab245440), VAV2 (ab52640), VAV3 (ab52938), GFP (ab290), actin (ab6276), calnexin (ab22595) were from Abcam. Other antibodies were, phospho-tyrosine (P5872, Merck), Rac1 (610650, BD Biosciences), TfR (13-6800, Thermo Fisher Scientific), Na/K-ATPase (NB300-146, Novus Biologicals) and Lamp1 (clone H4A3, Developmental Studies Hybridoma

Bank). Secondary antibodies: anti-human unconjugated (A18819) anti-human Alexa488 (A11013), anti-human Alexa568 (A21090), anti-human Alexa647 (A21445), anti-rabbit Alexa488 (A11008) and anti-mouse Alexa488 (A11001) were from Thermo Fisher Scientific. HRP-conjugated secondary antibodies: anti-mouse (172–1011) anti-human (172–1033) anti-rabbit (172–1019) and anti-goat (172–1034) were from BioRad.

Soluble TfR ectodomain (aa 89–760) was purified from baculovirus-infected Sf9 cell culture supernatants. After equilibration to 20 mM Tris pH 8 and 500 mM NaCl and filtration through a 0.22-μm membrane, the protein was purified by Ni-affinity purification using a HisTrapExcel column (17-3712-06, GE Healthcare), followed by size exclusion chromatography using a Superdex200 16/60 column running in 20 mM HEPES pH 7.5, 150 mM NaCl.

## Cell culture
Human cancer-derived cell lines: SkBr3 (ATCC HTB-30), HeLa (ECACC 93021013) and U2OS (ECACC 92022711) cells were grown in DMEM-GlutaMAX (Thermo Fisher Scientific 31966021) supplemented with 10% FBS, SkOv3 (ATCC HTB-770) cells were cultured in McCoy's 5 A modified medium (Thermo Fisher Scientific 16600082) supplemented with 10% FBS, and MCF7 (ATCC HTB-22) cells were grown in MEM (Thermo Fisher Scientific 41090028) containing 1× non-essential amino acids (Thermo Fisher Scientific 11140035) and 10% FBS. Chinese hamster ovary (CHO-K1) cells (ATCC CCL-61) were cultured in DMEM-GlutaMAX (Thermo Fisher Scientific 31966021) supplemented with, 10% FBS, and 0.3 mM proline (Merck 81709). Transduced CHO, HeLa and U2OS cells stably expressing HER2 full-length and HER2_ΔCT were cultured in the presence of 5 μg/ml blasticidin (Thermo Fisher Scientific A1113903). All mammalian cell lines were maintained at 37 °C in a 5% $CO_2$ environment. SkBr3 and U2OS VAV1-3 knockout (KO) cells were generated by transfection of cells with in vitro assembled ribonucleoprotein complexes (Synthego) using a Neon Electroporation system (Thermo Fisher Scientific MPK5000) according to the manufacturer's instructions. Guide RNAs were mixed at an equimolar ratio and assembled with Cas9 protein at 3:1 RNA:Cas9 ratio. Synthetic-modified guide RNA targeting sequences were as follows:

VAV1: UUCUAAUGUUCUUAAGGCAC, CUCACAGCAGGUGGACA GGA;

VAV2: AUCGUGGCAGACUUUCAGGA, GCCACGAUAAAUUUGGA UUA;

VAV3: UGUGUUUAUGGGGAAGAUGA, AUCUUCUUCAUCUUCCA CAA.

Single-cell clones were obtained by seeding cells onto 96-well plates 72 h post transfection by limiting dilution (one cell per well). Cell growth was monitored using an Incucyte imaging system (Sartorius) and verified single-cell clones propagated. VAV1-3 triple KO clones were identified by sequencing of genomic loci targeted by gRNAs. Sanger sequencing files were analyses using the Inference of CRISPR Edits (ICE) tool from ice.synthego.com. VAV2 KO was further confirmed by immunoblot analysis.

Spodoptera frugiperda (Sf9) cells (Thermo Fisher Scientific 11496015) were grown as suspension cultures in Insect-XPRESS medium (Lonza BE12-730Q) at 28 °C.

All cell lines were routinely screened for mycoplasma contamination.

## Chemicals
Small-molecule inhibitors were used at the following concentration if not otherwise stated: 200 nM BafilomycinA1 (ApexBio A8627), 300 μM CK666 (Merck 182515), 50 μM Cycloheximide (Merck C7698), 10 μM CytochalasinD (Santa Cruz sc201442), 80 μM Dynasore (Santa Cruz sc214953), 25 μM EHT1864 (Tocris 3872), 50 μM 5-(N-ethyl-N-isopropyl) amiloride (EIPA) (Merck A3085), 3.4 μM Lapatinib (Merck SML2259), 100 μM Ly294002 (Calbiochem 440202), 500 nM ML141 (Calbiochem

217708). TCEP (Merck 646547) was used at 50 mM for flow cytometry and 20 mM for confocal microscopy to quench cell-surface-bound DyLight-650-labelled ligands. Coomassie staining solution for in gel visualisation of proteins was InstantBlue (ISB1L, Expedeon).

## DNA constructs
Plasmids for transient transfection of GFP only pEGFP-N1 (Clontech) or GFP-N-terminal fusions to AP180ct (rat aa 530-915) and dynamin1_S45N have been described[9,11]. GFP-GPI was created by cloning the CD55 GPI-anchor sequence (aa 345-381) into pEGFP-N1. VAV1-3 C-terminally HA-tagged expression plasmids were from Addgene (#14553, #14554, #14554). All constructs described below were made by Gateway recombination (Thermo Fisher Scientific). HER2 constructs are based on cDNA reference sequence (NM_004448.4). Full-length HER2 (aa 1–1255) and a construct lacking the C-terminal cytoplasmic domain (aa 1–694) designated HER2_ΔCT, were C-terminally fused with the por-cine teschovirus-1-derived P2A cleavage sequence (ATNFSLLKQAGD-VEENPGP) followed by the blasticidin resistance cassette. All HER2-based constructs were recombined into a modified version of the pLenti-6-V5_Dest (Thermo Fisher Scientific V49610) in which the blasticidin resistance cassette had been exchange to puromycin. Trastuzumab non-binding mutants of HER2 (Tzmut) were engineered by deletion of aa 556-651 as previously described[15]. VAV2 lacking the SH2 domain (aa 665–772) was generated by PCR mutagenesis of the VAV2 ORF amplified from addgene plasmid #14554. VAV2 dominant-negative constructs comprising residues 546–878 or lacking catalytic residues 341–347 were generated based on previously published data on VAV1[32], by PCR mutagenesis of the VAV2 ORF amplified from Addgene plasmid #14554. Wt and mutant VAV2 constructs were recombined into a C-terminal monomeric-eGFP expression vector. Rac1 wt and dominant-negative mutant T17N as well as Ras dominant-negative mutant T17N were synthesised as codon-optimised gene-blocks (Integrated DNA Technologies), based on cDNA reference sequences (NM_006908.5, Rac1 and NM_005343.4 hRas iso1) and recombined into an N-terminal monomeric-eGFP expression vector. The transferrin-receptor ectodomain (aa 89–760) was cloned into a baculovirus expression vector containing an N-terminal melittin signal peptide sequence (MKFLVNVALVFMVVYISYIYAA) and a C-terminal 3 C cleavage site followed by a 10x-His tag. All plasmids were verified by DNA sequencing.

## Antibody and ligand uptake assays
Cells were serum starved for 1 h and pre-incubated with inhibitors for 10 min if not otherwise stated. All transient transfections were carried out 16–24 h before starting the assay. DyLight-650-labelled antibodies, were incubated at 3 μg/ml with cells for 30 min on ice and the excess was washed away with PBS. When secondary antibodies were used, those were subsequently added at 1.5 μg/ml for 30 min on ice and unbound material was removed by a PBS wash. Cells in serum-free medium (containing drugs as indicated) were incubated at 37 °C for times as shown in the respective experiment. Transferrin-546 was added at 10 μg/ml for the last 15 min of incubation time.

For dextran uptake, cells were incubated with 500 μg/ml Dextran (70 kDa)-Tetramethylrhodamine or Dextran (70 kDa)-Fluorescein ± DyLight-650-labelled BS4 or anti-TfR antibody 289 at 3 μg/ml for 10–30 min at 37 °C as specified in the figure legends.

DyLight-650-labelled WGA and biotin-transferrin were incubated with cells at 2.5 μg/ml for 30 min on ice and excess washed away with PBS. In the latter case, cells were subsequently incubated with tetra-valent streptavidin (Merck S4762) at 10 μg/ml for 30 min on ice and unbound material was removed by a PBS wash. Cells were incubated at 37 °C for 30 min to allow ligand uptake.

Antibody/ligand uptake was stopped by fixation in 4% PFA at 23 °C (on ice in case of surface labelling) for 10 min and cells were washed in PBS.

For analysis by flow cytometry, cells were detached using trypsin, pelleted, resuspended in 100 mM Tris pH 8.5 150 mM NaCl and 50 mM TCEP and immediately examined on a LSRFortessa cell analyser using FACSDiva V6.2 (BD Biosciences). A minimum of 10,000 cells were assessed with laser lines appropriate to the fluorophores used. Initial gating was area vs height to eliminate doublets and gating on forward and side scatter to exclude debris. Cells selected by this strategy were further gated on EGFP if transfected, providing intrinsic untransfected controls for each cell population. Transferrin and BS4 uptake was measured in the red and far-red channels, respectively both in untransfected (GFP-) and transfected (GFP + ) cells. The median of the cell populations was plotted and each data point shows the mean of three independent cell populations from three independent experiments.

For analysis by confocal microscopy, cells on glass coverslips were mounted in PVA-Dabco (Merck 10981), containing 20 mM TCEP (Merck 646547) to quench extracellular DyLight-650 signal, as indicated in the respective experiment. Alternatively, surface-bound antibody was stained by incubation with 2 µg/ml anti-human fluorescent antibody before glass coverslips were mounted in PVA-Dabco. Samples were analysed on a Zeiss LSM 780 or LSM 900 inverted confocal microscope equipped with a 63×/1.4NA objective. Images were acquired with identical settings among different experimental conditions using the ZEN software package (V2011 SP7 & V3.5 Blue edition, for LSM 780 and LSM 900, respectively), and post-processed using FIJI V2.3.0 (http://imagej.nih.gov/ij) with equal post-processing settings among different samples within an experiment. Subtraction of surface-bound from total antibody signal yielded the endocytosed pool, which was quantified as integrated density in individual cells. Dextran uptake was measured by quantification of intracellular fluorescence after background subtraction as integrated density in individual cells.

For continuous monitoring of antibody uptake, SkBr3 cells seeded onto 96-well plates were incubated with FabFluor pHRed (Sartorius 4722)-complexed antibody at equimolar ratio at 10 nM final concentration and immediately imaged every 2 min in the phase and red channel for the time of the experiment (1–16 h) using an Incucyte S3 Live-Cell Analysis System equipped with a ×20 objective (Sartorius). Each experimental condition was run in duplicate wells and two areas per well were imaged. Drugs were added or removed at indicated times by replacing the complete supernatant, and imaging continued without delay. Time-lapse images were analysed using the IncuCyte S3 Software package (V2018B, Sartorius) by quantifying the amount of red signal (endocytosed Ab-pHrodo conjugate) per cell area (phase images). Data from three independent experiments, normalised to the control condition are displayed.

## Live-cell microscopy

Cells grown on 35-mm glass bottom dishes (MatTek P35G-1.5-20-C) were transfected with GFP-GPI 24 h prior to the start of the experiment. Dishes were mounted and equilibrated at least 30 min before the start of the experiment on the microscope stage at 37 °C in a 5% $CO_2$ environment. DyLight-650-labelled BS4 antibody was added at 3 µg/ml and the imaging started immediately. Z-stacks of 300 nm increment steps covering a range of 3.6 µm were recorded every 10 s for the imaging period of 30 min using a Nikon iSIM microscope equipped with a 100×/1.49NA objective. Images were deconvolved using the NIS Elements AR (V5.11.01) software package (Nikon), and time-lapse 3D depth-colour-coded movies were generated using the same software. Surface rendering of the plasma membrane was done using Imaris (V9.7.2 ×64, Oxford Instruments). Maximum intensity Z and time projections were done in FIJI V2.3.0 (http://imagej.nih.gov/ij).

## Immunofluorescence

Cells grown on glass coverslips were incubated with antibodies in serum-free medium as indicated in the respective experiments and fixed for 10 min in 4% PFA at 23 °C. After a PBS wash, cells were permeabilised in 0.3% Triton in PBS and blocked in 5% goat serum, 5% BSA. Cell sections were incubated with primary antibodies, HER2 (CST 2242), HER3 (CST D2245) and EGFR (CST 4267), TfR (Thermo Fisher Scientific 13-6800), Na/K-ATPase (Novus Biologicals NB300-146) and Lamp1 (Developmental Studies Hybridoma Bank clone H4A3), all diluted 1:100 in 1% goat serum 1% BSA, for 1 h at 23 °C, followed by 3×10 min washes in PBS. Coverslips were incubated with 2 µg/ml (1:1000) corresponding secondary antibodies anti-rabbit Alexa488 (Thermo Fisher Scientific A11008), anti-mouse Alexa488 (Thermo Fisher Scientific A11001), or anti-mouse Alexa568 (Thermo Fisher Scientific A11004) for 1 h at 23 °C, and subsequently washed for 3×10 min in PBS. Cell sections were mounted in PVA-Dabco (Merck 10981) and analysed on a Zeiss 780 inverted confocal microscope equipped with a 63x/1.4NA objective. Images were acquired with identical settings among different experimental conditions using the ZEN software package (V2011 SP7, Zeiss), and processed using Fiji. Spot co-localisation was performed using a custom macro in Fiji. In brief, after background subtraction and threshold application (same values used for all experimental conditions), signal and reference channels were segmented followed by counting of overlapping spots. The macro is in the Source data file.

## Immunoprecipitation

SkBr3 cells grown on six-well dishes were incubated in a serum-free medium for 1 h with inhibitors. BS4 was added for 2.5 min at 3 µg/ml and after washing on ice with 1× PBS, cells were scraped in IP buffer (1× PBS, 1% NP-40, protease inhibitors (Roche 59813300), phosphatase inhibitors (Merck P5726)). After 15 min shaking at 4 °C, lysates were incubated with ProteinG Dynabeads (Thermo Fisher Scientific 10004D) (centrifugal clarification of lysates was avoided given that antibody-receptor aggregates readily spin down). After rotating beads in cell lysates for 2 h at 4 °C, beads were washed five time in IP buffer using a magnetic rack. After the final wash, beads were eluted in 1× Laemmli sample buffer and samples were analysed by western blot.

## Western blot

Cell lysates in 1× Laemmli sample buffer (10 mM Tris pH 6.8; 8% glycerol; 100 mM DTT; 2% SDS) were run on 4–12% Bis-Tris NuPAGE gels (Thermo Fisher Scientific NP0322) in 50 mM Tris; 50 mM MES; 0.1% SDS and blotted onto nitrocellulose membranes for 4 h at 4 °C in 25 mM Tris, 200 mM glycine, 25% (w/w) methanol. After blocking for 1 h at 23 °C with 5% milk in TBST (10 mM Tris pH 8; 150 mM NaCl, 0.05% Tween 20), membranes were incubated with primary antibodies diluted 1:1000 for 4–16 h at 4 °C, washed three times for at least 15 min in TBST, incubated with secondary HRP-conjugated antibodies diluted 1:1000 in 10% goat serum for 1 h at 23 °C and after 3 additional washes in TBST, developed using a chemiluminescent reagent (GE Healthcare RPN2232) in a Gel Doc XR + imager (Biorad 1708195). Uncropped blots and unprocessed scans are provided in the Source Data file.

## Receptor aggregation

Cells grown on six-well plates were incubated with antibodies/ligands in serum-free medium as specified. After washing the cells in PBS they were lysed in 0.5 ml of PBS + 0.2% Tx100 + Benzonase for 5 min on ice, insoluble material was separated by spinning the samples for 10 min at $20,000 \times g$ at 4 °C. Pellets were resuspended in PBS + 0.1% Tx100 and 0.1% SDS and sonicated before the addition of Laemmli sample buffer. Equal volumes of supernatant and pellet fractions were analysed by western blot. In total, 0.1% SDS was added to lysed cells and after sonication Laemmli sample buffer was added. Aggregation assays of recombinant proteins by WGA were performed using 30 µM of each protein as indicated in 10 µl final volume. Antibody-mediated aggregation of the TfR ectodomain was done in a 10 µl reaction containing

10 μM TfR (aa 89–760), 1.5 μM anti-TfR Ab 289, and 0.75 μM anti-human secondary antibody.

After incubation of the samples for 15 min at 23 °C, insoluble material was precipitated by a 5 min spin at 20,000 × g, 23 °C. Equal fractions of supernatant and pellet samples were analysed by Coomassie staining.

### Biotinyation of surface proteins and uptake assays

Surface biotinylation: Cells were washed twice with cold PBS and incubated 0.3 mg/ml EZ-Link NHS-biotin (Thermo Scientific 21328) in PBS for 1 h on ice. Remaining biotin was quenched with 50 mM Tris-PBS pH 8. After lysis and sonication (see aggregation assay above) lysate was incubated with Neutravidin agarose resin (Thermo Scientific 29204) for 50 min at 4 °C. Beads were washed and eluted with Laemmli buffer.

Endocytosis assays: Cell were labelled as above and returned to medium + 1% BSA at 37 °C for the period of the endocytic assay. Cell were returned to ice and surface biotin was removed 50 mM MESNa (Sigma M1511) in PBS for 2× 10 min. Excess MESNa was quenched with cold 20 mM Iodoacetamide (in PBS) for 5 min. Cells were washed with PBS and lysed as described above. A variation on this was where aggregates were pelleted, and after solubilized+ sonication, biotinylated protein (that had been aggregated) were isolated with Neutravidin agarose resin.

### Statistics and reproducibility

Numerical data were organised in tables using Excel (V15.24, Microsoft) and analysed using Prism (V8.4.3 & V9, Graphpad). Statistical analyses were performed using a two-tailed unpaired Student's $t$ test for pairwise comparisons. Multiple comparisons were done using one-way or two-way ANOVA with Dunnett's, Sydak's, or Tukey's multiple comparison test (as indicated in the figure legends), all included within Prism (V8.4.3 & V9, Graphpad). All of the assays were performed at least three times with similar results. The exact value of $n$, representing the number of biological replica, is indicated in the figure legends. Statistical differences with a $P$ value of 0.05 or less were considered significant. ns (non-significant) $P > 0.05$, $*P < 0.05$, $**P < 0.01$, $***P < 0.001$, $****P < 0.0001$.

### Reporting summary

Further information on research design is available in the Nature Portfolio Reporting Summary linked to this article.

## Data availability

Expression constructs of human proteins are based on the following reference sequences available at https://www.ncbi.nlm.nih.gov/nucleotide/: HER2 NM_004448.4, Rac1 NM_006908.5, Ras1 NM_005343.4. VAV1-3 expression data in Supplementary Fig. 8i is from proteinatlas.org: VAV1 ENSG00000141968, VAV2 ENSG00000160293, VAV3 ENSG00000134215. The authors declare that all other data supporting the findings of this study are available within the paper and its supplementary information files. Source data are provided with this paper.

## Code availability

The custom macro used in Fiji to perform spot co-localisation is available in the Source data file.

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

## Acknowledgements

We are grateful for invaluable support from MRC LMB scientific facilities: N. Barry and team (microscopy), M. Daly and team (flow cytometry). The authors gratefully acknowledge Jerome Boulanger for the Spot_Coloc macro. We wish to thank M.S. Bretscher, R.S. Hegde, R. Kay, and members of the McMahon lab for critical discussions and advice. We would like to thank the AstraZeneca colleagues P. Cariuk, R. Fleming, and Biologics Expression Team for generating the antibody molecules. This work was supported by the Medical Research Council, as part of United Kingdom Research and Innovation (also known as UK Research and Innovation) MRC file reference number U105178795 to H.M.M. For the purpose of open access, the MRC Laboratory of Molecular Biology has applied a CC BY public copyright licence to any Author Accepted Manuscript version arising. This project is supported through a research collaboration between AstraZeneca UK Limited and the Medical Research Council, Blue Sky program (project BSF10 to AB and HMM). The project was further supported by EMBO LTF 2015-1380 to D.P.

## Author contributions

D.P.: conceptualisation, methodology, investigation, writing—original draft, writing—review and editing, visualisation and funding acquisition. O.S.: conceptualisation, methodology, investigation and funding acquisition. Y.V.: methodology, validation, investigation and writing—review and editing. J.D.: resources, verification and project administration. A.B.: conceptualisation, writing—review & editing, project administration and funding acquisition. H.M.: conceptualisation, methodology, investigation, writing—original draft, writing—review and editing, visualisation, supervision, project administration and funding acquisition.

## Competing interests

A.B. and J.D. are employees of the AstraZeneca group of companies and have stocks/stock options in AstraZeneca. The remaining authors declare no competing interests.
