## [Peer Review File · Nature Communications]

Cell surface protein aggregation triggers endocytosis to maintain plasma membrane proteostasisEditorial Note: This manuscript has been previously reviewed at another journal that is not operating a transparent peer review scheme. This document only contains reviewer comments and rebuttal letters for versions considered at *Nature Communications*.

REVIEWERS' COMMENTS

Reviewer #1 (Remarks to the Author):

In their revised manuscript the authors have included some new experiments, including Rabankyrin-5 staining, EIPA inhibition, BafA effect, and signalling downstream of clustered HER2. This has improved the technical quality of the manuscript, but their rebuttal on the novelty of the findings is not compelling. In addition, it is difficult to understand the “general principle” the authors are referring to if there is no general mechanism for activation of the ADE pathway.

Reviewer #3 (Remarks to the Author):

The revised manuscript adequately addresses the concerns listed in my earlier review.

Revision 2

Reviewer #1 (Remarks to the Author):

In their revised manuscript the authors have included some new experiments, including Rabankyrin-5 staining, EIPA inhibition, BafA effect, and signalling downstream of clustered HER2. This has improved the technical quality of the manuscript, but their rebuttal on the novelty of the findings is not compelling. In addition, it is difficult to understand the "general principle" the authors are referring to if there is no general mechanism for activation of the ADE pathway.

While we welcome the reviewer's appreciation of the efforts that have been made to improve the manuscript, we disagree with his evaluation concerning the novelty (1) and general mechanism (2) described in our work.

We have answered all the queries thoroughly and did experiments that we did not think were necessary for the paper, but these were done to address all queries thoroughly (no doubt the paper was also improved). Now we find that the reviewer does not accept (1). the novelty: No one has ever previously suggested - let alone shown - that cells have a proteostatic pathway that can remove aggregates (whether miss-folded proteins or foreign bodies that bind or accumulate on the outside of the plasma membrane). One can say one does not accept the novelty but it would be better to back up the statements by showing where someone has ever shown this before.

Secondly (2) the reviewer questions whether there is a general mechanism. The general mechanism to us is that aggregation of receptors is a signal for their internalization via a Rac-actin dependent endocytic pathway. We have shown that HER2 receptors activate VAV but this does not mean that all other receptor will use the same GEF for Rac. There is a robustness in biology, and I think one is in danger of oversimplifying if one thinks that somehow all receptors will use the same adaptor proteins. We reject the referees claims and state strongly that we have indeed provided strong evidence of a Rac-actin-dependent endocytosis of HER2 and transferrin receptors and a variety of other receptors when aggregated, and we have shown that aggregation is the signal.

Reviewer #3 (Remarks to the Author):

The revised manuscript adequately addresses the concerns listed in my earlier review.

We thank the reviewer for the positive evaluation of the revised version of the manuscript.